# Lipidomics analysis of juveniles' blue mussels (*Mytilus edulis* L. 1758), a key economic and ecological species

Vincenzo Alessandro Laudicella[1]*, Christine Beveridge[1], Stefano Carboni[2], Sofia C. Franco[1], Mary K. Doherty[3], Nina Long[1], Elaine Mitchell[1], Michele S. Stanley[1], Phillip D. Whitfield[3], Adam D. Hughes[1]

1 Scottish Association for Marine Sciences, Dunstaffnage Marine Laboratory, Oban, United Kingdom,
2 Institute of Aquaculture, Faculty of Natural Sciences, University of Stirling, Stirling, United Kingdom,
3 Division of Biomedical Sciences, University of the Highlands and Islands, Centre for Health Sciences, Inverness, United Kingdom

* alessandro.laudicella@sams.ac.uk, alessandro.laudicella@gmail.com

**Data Availability Statement:** All relevant data are provided within the manuscript and its supporting information files. All raw data files are available

## Abstract

Blue mussels (*Mytilus edulis* L. 1758) are important components of coastal ecosystems and in the economy of rural and coastal areas. The understanding of their physiological processes at key life stages is important both within food production systems and in the management of wild populations. Lipids are crucial molecules for bivalve growth, but their diversity and roles have not been fully characterised. In this study, traditional lipid profiling techniques, such as fatty acid (FA) and lipid class analysis, are combined to untargeted lipidomics to elucidate the lipid metabolism in newly settled spat fed on a range of diets. The evaluated diets included single strains treatments (*Cylindrotheca fusiformis* CCAP 1017/2 – CYL, *Isochrysis galbana* CCAP 927/1– ISO, *Monodopsis subterranean* CCAP 848/1 – MONO, *Nannochloropsis oceanica* CCAP 849/10– NANNO) and a commercial algae paste (SP). Spat growth was influenced by the diets, which, according to their efficacy were ranked as follows: ISO>NANNO/CYL>SP>MONO. A higher triacylglycerols (TG) content, ranging from $4.23\pm0.82$ µg mg$_{ashfree\ Dry\ weight\ (DW)}^{-1}$ at the beginning of the trial (T0) to 51 $\pm15.3$ µg mg$_{ashfreeDW}^{-1}$ in ISO, characterised significant growth in the spat, whereas, a reduction of TG ($0.3\pm0.08$ µg mg$_{ashfreeDW}^{-1}$ in MONO), mono unsaturated FA–MUFA (from $8.52\pm1.02$ µg mgFA$_{ashfreeDW}^{-1}$ at T0 to $2.81\pm1.02$ µg mgFA$_{ashfreeDW}^{-1}$ in MONO) and polyunsaturated FA–PUFA (from $17.57\pm2.24$ µg mgFA$_{ashfreeDW}^{-1}$ at T0 to $6.19\pm2.49$ µg mgFA$_{ashfreeDW}^{-1}$ in MONO) content characterised poor performing groups. Untargeted lipidomics evidenced how the availability of dietary essential PUFA did not influence only neutral lipids but also the membrane lipids, with changes in lipid molecular species in relation to the essential PUFA provided via the diet. Such changes have the potential to affect spat production cycle and their ability to respond to the surrounding environment. This study evidenced the advantages of coupling different lipid analysis techniques, as each technique disclosed relevant information on nutritional requirements of *M. edulis* juveniles, expanding the existing knowledge on the physiology of this important species.

from the Mendeley data repository database (DOI: 10.17632/w57zy87s68.1).

**Funding:** This study was funded by the European Social Fund and Scottish Funding Council as part of Developing Scotland's Workforce in the Scotland 2014-2020 European Structural and Investment Fund Programme (Ref: UHI_SAMS_DSW_PGR_AY16/17). The work was also funded through SAICHatch project (MP_2015_02) funded by the Scottish Aquaculture Innovation Centre (SAIC, www.scottishaquaculture. com).

**Competing interests:** The authors have declared that no competing interests exist.

# 1 Introduction

Blue mussels are ecologically and economically important species, providing a range of crucial ecosystem services along with playing an important role in the economy of many rural and coastal regions [1]. The nutritional value of bivalves is also well documented, as they constitute a source of protein, amino acids, vitamins, trace metals and poly-unsaturated fatty acids– PUFA [2, 3]. Globally, bivalve production is important for food security, accounting for over 15.5% of total aquaculture production in 2016 [4]. Being passive feeders, bivalves reduce the nutrient load in the water [5] whilst not requiring the use of feeds for growth (as observed in intensive culturing of finfish and shrimps [6]), such characteristic makes bivalve culture an environmentally sustainable solution to future food security scenarios [4, 7–9]. Beside their role in bioassimilation of nutrients, mussels act as a filter for viruses, bacteria, detritus and phytoplankton [10], and as ecosystem engineers, creating shelter and substrate for other benthic organisms and increasing spatial complexity and biodiversity [11]. Furthermore, mussels are classic model organisms in ecotoxicological studies as they are sessile, ubiquitous in the marine environment and their filter-feeding behaviour mechanism [12, 13].

European mussel production relies uniquely on natural recruitment events, defined "spat-falls". Yet, due to influences of environmental drivers [14–16], spatfalls are subjected to severe yearly fluctuations. Such Irregular recruitment, alongside diseases, water quality classification and site licensing, are considered factors that are preventing the expansion of European mussel production [17, 18]. The establishment of commercial mussel hatcheries is a way to overcome some of these issues. Hatcheries can provide a continuous supply of juveniles to growers, resolving spatfall issues [19]. Nursery operations are a key stage for hatchery production of mussels, as the survival of mussel spat on-ropes depends on the size of spat [20–23]. Rearing newly settled mussel postlarvae (spat) in nurseries has significant effects on their survival when seeded on grow-out ropes, as it permits to reach a minimal size that ensures higher resistance towards the external environment [19]. However, feeding large amounts of spat in nurseries is expensive, particularly in term of diet supply, as algae production alone accounts around 40% of hatchery costs for rearing bivalve juveniles [24]. Furthermore, juvenile stages are characterised by high production losses, which happen when the hatchery products have their greatest value. Therefore, elucidating the physiological and nutritional requirmentas of mussel juveniles gain a great importance for the development of industrial hatchery production of mussels and of mussel aquaculture production.

Early studies recognised lipids to be the main energy stores for bivalves up to 6 months post-settlement [25, 26]. Since then, evaluating the lipid composition of diets and juveniles gained a central importance on bivalve nutrition studies with examples available for clams [27–33], scallops [34–36], oysters [37–42] and mussels [19, 43–46]. Other than protein and carbohydrate composition, the nutritional properties of shellfish diets strongly depend on the essential PUFA (arachidonic acid– 20:4n-6, AA; eicosapentaenoic acid– 20:5n-3, EPA; docosahexaenoic acid– 22:6n-3, DHA) content [47]. In spite of their ability to *de novo* synthesize lipids, bivalves have limited ability to elongate and desaturate C18 fatty acids (FA) to essential PUFA [27, 29, 48, 49]. FA desaturation requires energy; which could, in turn, penalises growth [30]. To date, a large part of the knowledge on lipids composition and roles in bivalve juvenile nutrition is based on FA analysis. However, this technique only captures a proportion of the lipidome, and lipids have a key role in organism physiology, ranging from biological membranes to cell signalling, immune responses and energy reserves [50]. Current advances in technologies and analytical platforms allow for a deeper and global analysis of lipid molecular species, known as lipidomics. Lipidomics, a branch of metabolomics, encompasses the totality of biological lipids (the lipidome) of a living organism [51]. Lipidomics is mainly based on

liquid chromatography coupled with mass spectrometry (LC-MS) platforms [52]. By working with liquid chromatography, information are obtained at lipid molecular species level, rather than on lipid sub-fractions (e.g. fatty acid) or lipid classes. Through lipidomics, global changes on lipidome can be visualised, obtaining essential physiological information of the examined organisms [53].

The aim of this paper is to expand existing knowledge on lipid metabolism during the crucial post larval (spat) life stage. As such, we applied a comprehensive lipid analysis strategy, which included FA profiling, lipid class analysis and untargeted lipidomics, to evaluatate of the effects of four single strain microalgae species and one commercial algae paste on newly settled *Mytilus edulis* L. 1758. To our knowledge, this is the first time that such holistic lipid analysis approach is applied to bivalve juveniles.

## 2 Materials and methods

### 2.1 Research ethics

This project obtained ethical approval from the University of the Highlands and Islands with code ETH884 Lipidomics and Proteomics investigation of Commercial Bivalve production in Scotland.

### 2.2 Spat collection and experimental design

Newly settled *M. edulis* (shell length <10 mm) were obtained from Inverlussa Marine Services (www.inverlussa.com; Isle of Mull, West Coast of Scotland) in July 2017. Spat were collected via gentle scraping of spat collectors, bagged in plastic bags filled with seawater and transferred in ice to the aquarium facilities of Scottish Association for Marine Science (SAMS). Upon arrival, the juveniles were graded onto a 4 mm mesh and then kept for 48 hrs on sand-filtered (Grade 1, EcoPure, Waterco) seawater, at room temperature with no food, to allow acclimation and depuration of the animals. After the period of acclimation, spat were sorted and divided into the experiment groups (shell length <5 mm). Groups of 10 selected spat were photographed on milli-graph paper (to obtain shell length–SL) and weighted to 0.0001 g (total live weight–LW). Each group of 10 individuals was successively placed into a 1.5 mm nylon mesh. Three groups of 10 spat constituted the experimental unit (N = 30). Three further lots of spat were deployed in an open water environment (OUT) and used as a reference for the laboratory feeding trial.

The feeding trial lasted for 28 days, during which the spat were placed in 8 L conical tanks, kept at 18°C in a static system with 20 μm filtered seawater and gentle aeration, under an 18:6 (Light:Darkness, L:D) photoperiod, with each diet treatment replicated in three independent tanks. The water was fully changed three times per week, coinciding with the feeding of the spat. Salinity (hand refractometer), pH (HI98190, Hanna instruments), dissolved oxygen (Fibre optic oxygen transmitter, PreSens) and ammonia (Test $NH_3/NH_4^+$, Tetra) were monitored before every water change for the entire feeding trial, temperature loggers (Pendant, HoboWare) were used to monitor continuously the temperature profile of the tanks.

Five diet treatments were evaluated during the trial, one of which included Shellfish Paste (SP–Instant Algae 1800, Reed Mariculture), which is a mixture of *Isochrysis* spp., *Pavlova* spp., *Tetraselmis* spp., *Chaetoceros calcitrans*, *Thalassiosira weissflogii* and *Thalassiosira pseudonana*. The remaining treatments included the administration of microalgae mono-diets of *Cylindrotheca fusiformis* (CYL–CCAP 1017/2), *Isochrysis galbana* (ISO—CCAP 927/1), *Monodopsis subterranean* (MONO–CCAP-848/1) and *Nannochloropsis oceanica* (NANNO–CCAP 849/10). All strains used in this study were provided by the Culture and Collection of Algae and Protozoans (CCAP, www.ccap.ac.uk). Diets were supplied at a weekly ration of 0.4 mg of diet

dry weight for each mg of live weight of reared spat [24]. Every week the spat were weighted and their live weight used to calculate the amount of diet to be provided during the following week. The clearance rate was monitored via a turbidimeter (TN-100, Eutech) both before and after diet administration, as an indicator of active grazing (**S1 Fig**). At the end of the trial, both OUT and laboratory kept spat were left for 48 hrs to depurate in filtered seawater and then snap-frozen in liquid nitrogen and stored at -80˚C for further analysis. Spat were then freeze-dried (18 LO plus, Christ) and ground to a fine powder in liquid nitrogen. Ash content was calculated following the combustion of powdered spat for 12 hrs at 450˚C.

## 2.3 Microalgae production

Microalgae were grown in sterile 2 L Duran's fitted with aeration lids, tubing and filters kept at 21˚C under a 16:8 L:D photoperiod. Media used to maintain each strain are reported in **Table 1**. To obtain the cell dry weight for each strain (needed to calculate the required weekly food ratio for the spat), 12 aliquots of 1 mL were collected from each stock culture; 6 of them counted using a coulter-counter (Multisizer 3, Coulter Counter) and the remaining 6 freeze-dried to obtain the dry weight which was then reported to the number of cells contained. Weekly, microalgae were harvested via centrifugation at 13G (9000 rpm) for 20 mins 4˚C using sterile 250 mL centrifuge tubes (VWR) and concentrated in a 50 mL sterile tubes (VWR) which were kept at 4˚C and used for feeding the spat. At every harvesting day, an aliquot (50 mL) from each strain was collected, centrifuged (14000 rpm 4˚C 10 min) and placed in 1.5 mL tube (Eppendorf). The tubes were snap-frozen, freeze-dried and kept at -80˚C for lipid analysis.

## 2.4 Biometrical analyses

Spat growth rate (GR) was measured as shell length increase (SI) and live weight increase (WI). At the beginning and at the end of the trial, spat were photographed on milli-graph paper, and their shell length (SL) was obtained by processing the images via ImageJ software (www.imagej.nih.gov). SI was calculated as Δ between shell size at end (T28) and the beginning of the feeding trial (T0). For WI calculation, spat were blotted on tissue paper and weighted at 0.0001 g scale (Sartorius). The Δ between the live-weight (LW) at T28 and T0 resulted in the WI.

## 2.5 Biochemical analyses

**2.5.1 Lipid extraction.** Microalgae and diet samples were resuspended in 200 μl of milliQ water and disrupted via probe sonication for 1 minute, whilst aliquots (≈ 10 mg) of powdered spat were homogenated in 200 μl of milliQ water by pestling in ice for 1 minute. For all samples, lipid extraction was done according to Folch, Lees [54], a detailed overview of sample preparation is provided in **S1 Protocol**. The dried lipid extracts were weighted to the 0.00001 g (Sartorius) and resuspended in 0.5 mL of chloroform constituting the total lipid extract (TLE). The TLE was divided into 2 sub-aliquots. One aliquot (400μl) was dried in nitrogen and stored at -80˚C for lipid class and lipidomics analysis. The second aliquot (100 μl) was spiked with an internal standard (FA 17:0, Sigma + 0.001% of BHT, Cayman Chemical Company at the 10% of the total lipid mass) and processed for fatty acid methyl esters (FAME) analysis.

**2.5.2 Fatty acids analysis.** FAME from TLE of diets and spat were prepared by acid-catalysed transesterification according to AOCS [55]. The FAME layer was evaporated under a gentle nitrogen stream (NVap, Organomation) and the FAME were resuspended in 500 μl of iso-hexane, purified on silica SPE cartridges (Clean-up Cusil 156, UCT) preconditioned with 5 mL of iso-hexane, and eluted twice with 5 mL of a 95:5 iso-hexane:diethyl ether solution. Purified

**Table 1. Summary of diets employed during the feeding trial.** For the live algae treatments, media used in the culture of microalgae strains and relative CCAP codes are reported.

| Strain CCAP code | Species name | Media | Diet code | Feeding group code |
|---|---|---|---|---|
| N/A | *Shellfish paste* | N/A | **ShellPaste** | SP |
| 1017/2 | *Cylindrotheca fusiformis* | F/2 + Si | *C. fusiformis* 1017/2 | CYL |
| 927/1 | *Isochrysis galbana* | F/2 | *I. galbana* 927/1 | ISO |
| 848/1 | *Monodopsis subterranean* | 3N BBM+V | *M. subterranean* 848/1 | MONO |
| 849/10 | *Nannochloropsis oceanica* | F/2 | *N. oceanica* 849/10 | NANNO |

FAME were dried in a N-vap and resuspended to 1 mg mL$^{-1}$ according to the original lipid mass in iso-hexane (HPLC grade, Fisher). To verify the endogenous abundance of FA17:0 in diets and spat tissues, FAME extracted from unspiked samples were also subjected to GC analysis. From the analysis of these samples, FA17:0 was observed only in trace in the diets samples, whereas endogenous FA17:0 was observed in spat tissues in a constant content, showing in average a ten-fold lower intensity than the internal standard peak.

FAME were separated by gas chromatography using a Trace GC Ultra (ThermoFisher) equipped with a fused silica capillary column (ZBWax, 60m x 0.32 x 0.25 mm i.d.; Phenomenex) with hydrogen as a carrier gas and using on-column injection. The temperature gradient was from 50 to 150˚C at 40˚C min$^{-1}$ and then to 195˚C at 1.5˚C min$^{-1}$ and finally to 220˚C at 2˚C min$^{-1}$. Individual methyl esters were identified by comparison to known standards (Marine Oil, Restek) and by reference to published data [56]. Identification of dimethylacetals and non methylene interrupted dienoic (NMID) FA, as well as further unknown peaks not included in commercial standard mixtures was done via GC-MS (Trace GC ultra combined with Trace DSQ, ThermoFisher). Data were collected and processed using the Chromcard for Windows (version 2.00) computer package (Thermoquest Italia S.p.A.). Results are reported as a relative percentage of FAME composition (%FAME) and as absolute quantification via the internal standard method (μglipid mg$_{DW}$$^{-1}$ for microalgae and μgFA mg$_{ashfreeDW}$$^{-1}$ in the case of spat).

**2.5.3 Lipid class analysis.** TLE from spat and diets were separated in their main lipid class via normal phase high-pressure liquid chromatography coupled with electron light scattering detector (NP-HPLC-ELSD). The separation was accomplished with an Infinity 1260 platform (Agilent Technologies) according to Graeve and Janssen [57] with minor modifications. The protocol was modified to enhance the separation of certain lipid classes relevant in marine invertebrates, such as phosphonoethyl ceramides (PE-Cer). TLE was separated on a monolithic silica column (Chromolith Si 100x4.6, Merck) equipped with the relative guard columns (Chromolith Si guard cartridges, Merck). The column was kept at 40˚C and the solvent flow kept at 1.4 mL min$^{-1}$. The quaternary elution gradient is reported in **Table 2**. Acetone, isooctane (tri-methyl pentane), ethyl acetate and water were all obtained from FisherBrand and HPLC grade. HPLC grade isopropanol (IPA) was obtained from Chromanorm. Glacial acetic acid (GAA) and triethylamine (TEA), both HPLC grade, were purchased from VWR.

Identification of principal lipid classes was achieved via an external standard method. Commercially available purified lipid fractions were used as lipid class standards. Squalene (terpenes—TER), Arachydil dodecanoate (wax ester–WE), Cholesterol (free sterols–ST), triglycerides (triolein–TG), diacylglycerols (DG), monoacylglycerols (MG), FA17:0 (free fatty acid–FFA), Ceramides lipid mix from bovine brain (Cer), Phosphatidic acid (PA), phosphatidyl ethanolamine from soybean (PE), Cardiolipin from bovine heart (CL), Phosphatidyl serine (PS), phosphatidyl choline (PC), phosphatidyl inositols (PI), lyso-phosphatidyl choline from egg yolk (LPC) all obtained by Sigma and sphingosylphosphorylethanolamine (PE-Cer) from

**Table 2. Quaternary gradient used during NP-HPLC separation of spat TLE.** Mob A: Isooctane:Ethyl Acetate (99.8:0.2); Mob B: Acetone: Ethyl Acetate (2:1) + 25 mM GAA; Mob C: IPA: Water (85:15) + 15mM GAA and 7.5 mM TEA; Mob D: Isopropanol.

| Ret. Time (min.) | Mobile phase | | | |
|---|---|---|---|---|
| | A(%) | B(%) | C(%) | D(%) |
| 0.0 | 100 | 0 | 0 | 0 |
| 0.1 | 100 | 0 | 0 | 0 |
| 1.5 | 100 | 0 | 0 | 0 |
| 1.6 | 97 | 3 | 0 | 0 |
| 6.0 | 94 | 6 | 0 | 0 |
| 8.0 | 50 | 50 | 15 | 0 |
| 8.1 | 46 | 39 | 24 | 0 |
| 14.0 | 43 | 30 | 24 | 0 |
| 14.1 | 43 | 30 | 60 | 0 |
| 18.0 | 40 | 0 | 60 | 0 |
| 23.0 | 40 | 0 | 0 | 0 |
| 23.1 | 0 | 100 | 0 | 0 |
| 25.0 | 0 | 100 | 0 | 0 |
| 25.1 | 0 | 0 | 0 | 100 |
| 27.0 | 0 | 0 | 0 | 100 |
| 30.0 | 100 | 0 | 0 | 0 |
| 32.0 | 100 | 0 | 0 | 0 |

Matreya. Stock solutions for each lipid were prepared at 2.5 mg mL$^{-1}$ in 2:1 chloroform:methanol (HPLC grade, Fisher). From the stock solution, working solution were diluted in Mob A. Calibration curves were calculated by sequential 10 µl injections of standard mix solutions (0.5–0.25–0.125–0.066–0.033–0.0165–0.008 µg µl$^{-1}$ of each lipid class). Identification of lipid classes was achieved by retention time match between unknown samples and the standard mix. Spat TLE were resuspended in a 4:0.06:0.04 (Mob A:chloroform:methanol) solution at a concentration of 1 mg mL$^{-1}$ of which 10 µL volume was injected. Chromatograms were inspected, integrated and calibration curves calculated via Chemstation software (Agilent Technologies). Results are reported as relative lipid class composition (%TLE) and in absolute values (µglipid mg$_{ash\ freeDW}$$^{-1}$).

**2.5.4 Untargeted lipidomics.** Untargeted lipidomics of spat was achieved via High-Resolution Mass spectrometry (HRMS). The platform used was a binary HPLC (Accela, ThermoFisher) coupled with an electron spray ionization (ESI) and orbitrap mass analyser (Exactive, ThermoFisher). The separation was done on a C18 Hypersyl Gold 100x2.1 mm 1.9nm particle size (ThermoFisher) kept at 50°C. The binary solvent system included a constant flow rate of 400 µL min$^{-1}$ with a gradient as described in Table 3. Water and acetonitrile were HPLC grade and obtained from Fisher, IPA was LC-MS grade (Hypergrade LiChrosolv, Merck), while ammonium formate and formic acid were both LC-MS grade and obtained from Sigma Aldrich.

The mass spectra were acquired in the m/z range 250–2000 both in positive ESI (POS) and in negative ESI (NEG) with a mass resolution power of 100,000 FWHM. The mass error was kept below 5 ppm by routinely calibrations on both polarities with a calibration solution (Pierce™ LTQ ESI calibration solutions, ThermoFisher). TLE from spat were resuspended in 3:1 methanol: chloroform at a concentration of 1 mg mL$^{-1}$ with 3 µL injection volume. Chromatograms and mass spectra were inspected and integrated via Xcalibur software

**Table 3. Binary gradient used during LC-MS analysis of spat TLE.** Mob A: Water + 10 mM Ammonium formate + 20 mM Formic acid; Mob B: IPA:ACN (9:1) + 10 mM ammonium formate + 20 mM formic acid.

| Ret. Time (min.) | Mobile phase | |
|---|---|---|
| | A(%) | B(%) |
| 0.0 | 65 | 35 |
| 0.5 | 65 | 35 |
| 4.0 | 35 | 65 |
| 19.0 | 0 | 100 |
| 21.0 | 0 | 100 |
| 21.1 | 65 | 35 |
| 27.0 | 65 | 35 |

(ThermoFisher), data processing and analysis procedures are reported in the data analysis section. LC-MS profiles of spat extracts are reported in **S2 Fig**. Features were identified according to their precursor ion exact mass (MS') and reported as lipid class with the total number of carbons and double bonds (e.g. PC(36:5), for phosphatidyl choline with 36 carbons and 5 double bonds on the fatty acyl residues). Isobaric lipids separated by reverse phase chromatography but evidencing same MS', are reported with different letters (e.g. PC(38:5)$_a$ PC(38:5)$_b$).

## 2.6 Data analysis

**2.6.1 Biometrical, FAME and lipid class analyses.** Statistical analysis was compiled via R statistical software (version 3.5.1). Data are reported as mean ± standard deviation (SD) unless differently stated. Statistical differences were considered significant for $p < 0.05$. Biometrical data were log-transformed to force homoscedasticity. If normality assumptions were met, a two-way analysis of variance (two-way ANOVA) and a Tukey HSD test were employed to evaluate differences between the different diet groups at each sampling point. If homoscedasticity, following data transformation, was not met a Kruskal-Wallis with a Dunn's test (R 'dunn.test' package) as *posthoc* test was used to evaluate the effects of diet treatments on the spat.

Lipid class and FAME data were square-root transformed and multivariate differences were evaluated via Analysis of Similarities (ANOSIM). Non-Metric Multidimensional Scaling (nMDS) with Euclidean distance matrix was employed to graphically plot each sample group. Similarity percentages (SIMPER) were applied to evaluate the main lipid and fatty acids characteristic for each groups clustering. ANOSIM, nMDS and Simper analysis were obtained from R 'vegan' package [58]. Lipid class and fame composition differences between groups were evaluated via a one-way ANOVA and Tukey HSD (false discovery rate–FDR–adjusted p-value [59]) as *posthoc* test, whilst Kruskal-Wallis with a Dunn's test as *a posteriori* comparison was employed on features that failed normality assumptions (tested via a Cochran test).

**2.6.2 Untargeted lipidomics.** Raw LC-MS data were processed via Progenesis QI software (Nonlinear Dynamics, Waters). A technical QC sample was run every 6 hours of instrument operation time to monitor possible technical shift in the machine. Chromatograms were automatically aligned using a QC as a reference point. Peak picking and deconvolution were completed following automatic settings of the software with a set intensity threshold of $1xE^5$ and $1xE^4$ respectively for POS and NEG ionization modes. Data were normalised according to the total ion current of each chromatogram. Main lipid adducts for both POS and NEG were experimentally evaluated by using a lipid standard mixture that included the main lipid classes and were added to the software search (**S1 Table**). Lipid identification was achieved by

searching the lipid dataset versus LIPID MAPS (www.lipidmaps.org), HMDB (www.hmdb.ca), Metlin (www.metlin.scripps.edu) and an "*in house*" bivalve lipid database built from recent bivalve lipidomics studies [60–69]. Contaminants were manually removed from the peak intensity table (PIT) generated from this process.

The PIT was furtherly filtered and processed with the R based package 'MetaboAnalystR' [70]. Filtering process included the removal of features with over 30% of missing values and substitution of remaining missing values with a small value (half of the minimum intensity value). Features with low repeatability or low constant values were filtered out using QC samples and inter-quantile range, data was then scaled via Pareto scaling, to reduce the skewness of data and enhance comparability between different samples. Chemometrics analysis was employed as data reduction and biomarker discovery tools. Principal component analysis (PCA) was used to evaluate data quality, clustering between QC samples and the presence of outliers (S3 Fig). Partial least squares discriminant analysis (PLS-DA) was applied to cluster samples and to calculate variables of importance in projection (VIP) scores, which represent the weighted sum of squares of the PLS loadings taking into account the amount of explained Y-variation [71]. PLS-DA model fitting was evaluated via a permutation test and ten-fold leave one out cross-validation (LOOCV, S4 Fig). The statistical significance of the features with VIP scores >1 was furtherly screened via a Kruskal-Wallis test with FDR adjusted p-value [59]. Significantly different VIP>1 features were subjected to hierarchical clustering (HC) analysis and reported as heatmap (plotted via the R package 'pheatmap' [72]). Number of significant clustering selected for data representation was identified via the "elbow method".

**2.6.3 Identification of main lipids linked with growth spat performances.** The correlation between spat growth performances (in term of WI) and spat lipid composition (considering the significant lipids evidenced from multivariate analysis of FAME, lipid class and lipidomics dataset) was calculated by means of Spearman rank correlation coefficient. Absolute values for lipid class and FAME data and relative intensity for lipidomics data were used to compute the correlation.

# 3 Results

## 3.1 Diets fatty acid (FA) composition

The diets employed in this study presented a distinct FA profile (Fig 1). The complete table for relative (%FA) and absolute ($\mu$gFA mg$_{DW}^{-1}$) composition of the diets employed in the trial is reported in S2 Table. Total saturated FA (SAFA) was highest in *C. fusiformis* and *I. galbana* (p<0.05), while the highest amount of monounsaturated FA (MUFA) was observed in *N. oceanica* (S2 Table). *C. fusiformis* resulted in the richest diet for n-6 PUFA (p<0.001) whilst no evident differences between diets were found for total n-3 and total PUFA content (p>0.05).

ANOSIM (R 0.998 p<0.001) evidenced the presence of multivariate differences between the various diet. *I. galbana* was characterised by a high content of 22C FA as 22:1n-9, 22:5n-6, 22:5n-3 and DHA (p<0.001, Fig 1A, 1B and 1C), while lacked EPA and AA. Further relevant FA in *I. galbana* resulted 14:0, 18:1n-9, 18:2n-6, 18:4n-3 and 20:2n-6. On the contrary *C. fusiformis* and *N. oceanica*, respectively resulted rich in AA (p<0.001) and EPA (more abundant in *N. oceanica*, p<0.01) and the MUFA 16:1n-7, which accounted for the 20% of total FA in both strains. *M. subterranean* was characterised by the richness of C18 FA as 18:2n-6, 18:3n-6 and 18:3n-3 (p<0.001). *N. oceanica* and *C. fusiformis* were poor sources of 18:4n-3, which accounted for the 5% of total FA in all the other diets (p<0.001).

ShellPaste, as a mixture of different microalgae strains, resulted in a balanced composition of main essential PUFA as EPA (16.1±0.75%/11.86±2.73 $\mu$gFA mg$_{DW}^{-1}$) and DHA (6.17% ±1.63%/4.47±1.25 $\mu$gFA mg$_{DW}^{-1}$), while lacked in AA. *I. galbana* presented DHA as main

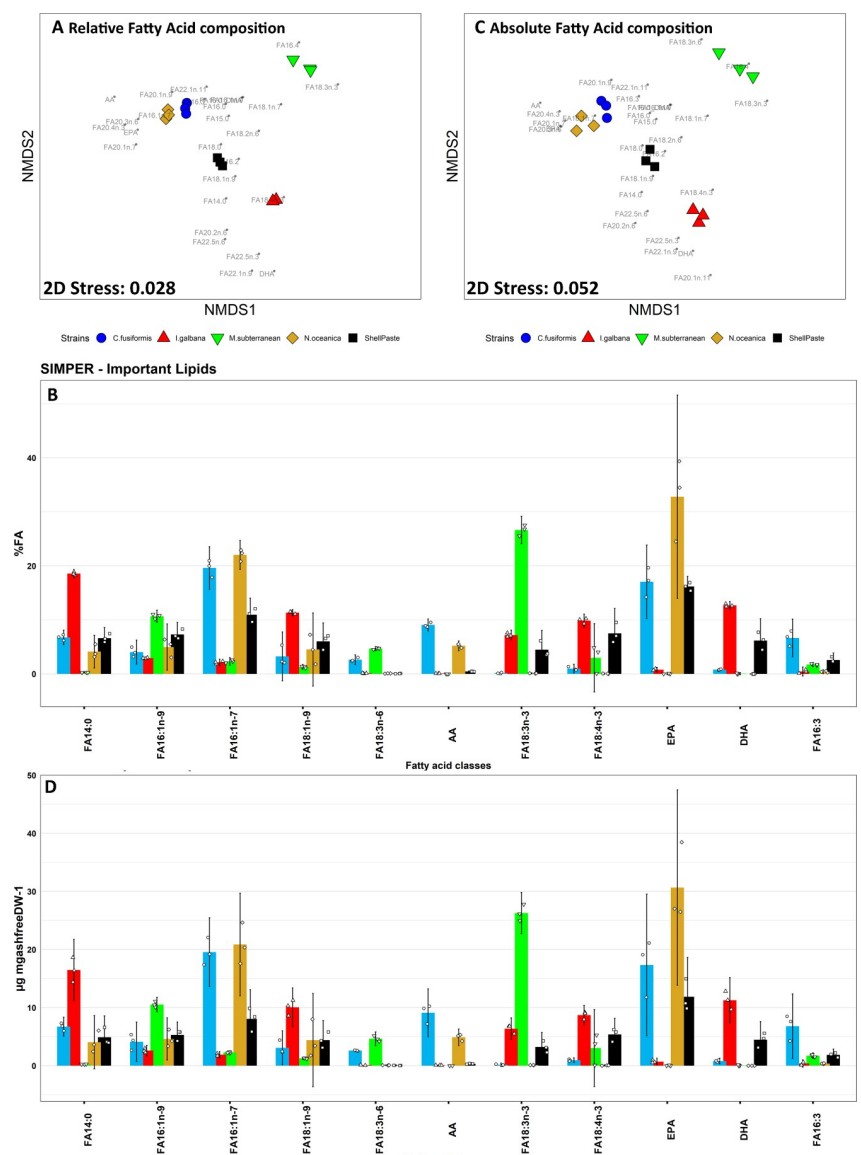

**Fig 1.** Non metric multidimensional scaling (nMDS) analisis and similarity percentages (SIMPER) analysis for relative (A-B) and absolute (C-D) fatty acid composition of diets. A: nMDS plot of the relative FA composition of diets (% total FA) provided during the feeding trial. Single FA loadings are stacked on the plot and reported in grey. B: principal variables explaining for a cumulative 75% of group differences in relative diet FA composition data evidenced by SIMPER analysis. C: nMDS plot of the absolute ($\mu$gFA $mg_{DW}^{-1}$) fatty acid composition of the diets. Single FA loadings are stacked on the plot and reported in grey. D: principal variables explaining for a cumulative 75% of group differences in absolute diet FA composition data evidenced by SIMPER analysis. Three replicates (n = 3) for each diet are here reported Charts B-D: data are reported as average (bar chart) ± 95% confidence interval; individual observations are jittered on the chart (smaller dots). Full data including all the FA observed and details on statistical significance values are found in **S2 Table**.

essential PUFA (12.69±0.24% /11.27±1.57 µgFA $mg_{DW}^{-1}$). AA and EPA were the most abundant PUFA in *C. fusiformis* and *N. oceanica*, with AA resulting higher in *C. fusiformis* than *N. oceanica* (respectively 9.05±0.44% / 9.11±1.66 µgFA $mg_{DW}^{-1}$ and 5.20±0.56%/4.90±0.57 µgFA $mg_{DW}^{-1}$) and with EPA showing the opposite trend (respectively 17.05±2.74%/17.34 ±4.91 µgFA $mg_{DW}^{-1}$ and 32.79±7.57%/30.67±6.77 µgFA $mg_{DW}^{-1}$). On the other hand, *M. subterranean* lacked in C20 PUFA and was characterised by large amounts of 18:3n-3 (26.63 ±1.02%/26.28±1.42 µgFA $mg_{DW}^{-1}$).

### 3.2 Spat growth performances during the feeding trial

At the end of the feeding trial period, we observed that the GR in spat largely varied across the sample groups (**Fig 2**). After 4 weeks of deployment at sea, OUT resulted the group with the longest SL (4.86±0.68 mm) and highest LW (75.08±14.2 $mg_{LW}spat^{-1}$). Although smaller than OUT, also ISO, NANNO, CYL and SP resulted in bigger shells compared with the beginning of the trial (p< 0.001, **Fig 2A**), whereas MONO did not present any significant increase in their SL or LW. On the contrary, spat LW significantly increased in ISO, CYL and SP treatments (Although SP resulted still smaller than the other groups, **Fig 2C**). Nevertheless, considering SI and WI, ISO and OUT outperformed the remaining sample groups (p<0.01, **Fig 2B–2D**).

### 3.3 Fatty acid composition and lipid class analysis of spat

The complete fatty acid (FA) composition and lipid class data, expressed as relative composition (%) and absolute value ($μg\ mg_{ashfreeDW}^{-1}$) of FA and lipid classes, are reported in **S3 Table**. nMDS plots and SIMPER important values identified from FA composition and lipid class composition analysis are reported respectively in **Figs 3** and **4**. ANOSIM analysis suggested the presence of strong multivariate differences both for FA and lipid class between sample groups (R 0.904 p <0.001 for FA and R 0.707 p<0.001 for lipid class).

Considering the nMDS plots provided in **Fig 3A and 3C**, T0 is located in the center of the plots, and from there three main cluster groups are observed (SP+MONO, CYL+NANNO and ISO+OUT) according to the relative (%FA) and absolute ($μgFA\ mg_{ashfreeDW}^{-1}$) FA compositon of the spat. OUT and ISO are characterised by the content in 18:2n-6 (p<0.05), 18:3n-3 (P<0.01) and 18:4n-3 (p<0.001, **Fig 3B–3D**). ISO also contained larger amounts of 14:0 (p<0.001), 18:1n-9 (p<0.01), of the PUFAs 20:2n-6 (p<0.05) and 22:5n-6 (p<0.05). EPA was significantly lower in ISO than T0, CYL, NANNO and OUT (p<0.01). A Second group cluster was formed by CYL and NANNO, characterised by the higher amount of 16:1n-7 (p<0.05), AA (p<0.05) and by the low DHA content (p<0.01). CYL also evidenced the presence of 22:4n-6 and 22:5n-6 (p<0.05), whilst higher levels of 18:1n-9 were observed in NANNO (p<0.05). A reduction of 22:2 and 22:3 non-methylene interrupted dienoic (NMID) FA was also observed in CYL and ISO (p<0.05). The third cluster included SP and MONO characterised by low EPA (p<0.01) and their high relative content of 16:4, 20:2 and 22:2 NMID (p<0.01). DHA content was significantly lower in SP compared with T0 (p<0.05).

The relative content for SAFA resulted lower in ISO than T0 (**S3 Table**, p<0.05), whereas NANNO was characterised by the highest relative content of MUFA (p<0.001). On the other hand, the absolute content of both parameters resulted significantly lower in SP and MONO when related with the remaining sample groups (**S3 Table,** p<0.05). The relative content n-6 PUFA resulted higher in all laboratory reared spat groups (p<0.05, **S3 Table**), whilst considering the absolute content of n-6 PUFA these were accumulated only in CYL and ISO (p<0.05). Other important differences were observed in the absolute content of n-3 PUFA (p<0.01) and total PUFA (p<0.05), which decreased significantly in MONO and SP in comparison with T0.

Lipid class composition analysis evidenced the principal lipid classes found in spat which included terpenes (TER), wax esters (WE), triacylglycerols (TG), free sterols (ST), diacylglycerols (DG), free fatty acids (FFA), phosphatidylethanolamines (PE), phosphatidylinositols (PI), phosphatidylcholines (PC) and lysophosphatidylcholines (LPC). The nMDS plot evidenced the presence of three main groups of samples according to the relative lipid class composition of spat subjected to the different dietary treatments (**Fig 4A**). T0 and SP contained similar relative amounts of neutral and polar lipids (**S3 Table**) and occupied the central portion of the nMDS plot. A large percentage of neutral lipids in these samples were constituted by TER

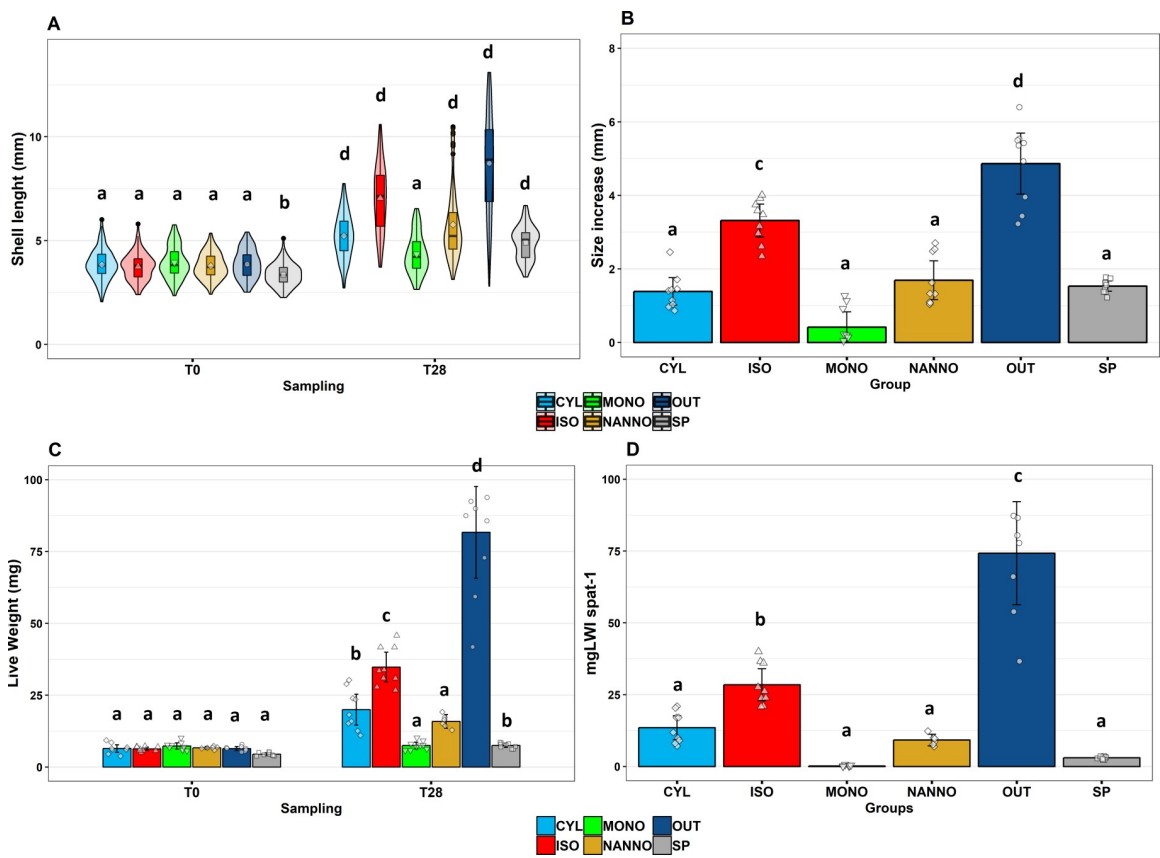

**Fig 2. Spat growth performances when subjected to the different diet treatments.** A: variation of shell length in mm between in each group between the beginning of the trial (T0) and end of the experiment (T28). B: Size increase of the spat (reported in mm) during the diet trial when subjected to different diets. C: live weight per spat (reported in mg) between the beginning of the trial (T0) and the end of the trial (T28 D: weight increase per spat throughout the trial according to the different sample groups (reported in mg). SP: spat subjected to Shellfish Paste, CYL: spat subjected to *C. fusiformis* 1017/2, ISO: spat subjected to *I. galbana* 927/1, MONO: spat subjected to *M. subterranean* 848/1, NANNO: spat subjected to *N. oceanica* 849/10, OUT: Outdoor deployed spat. A: N = 90 individual spat per group, marker in each box plot indicates the group average, band inside the box indicates the group median, box includes observation between 1st (25th percentile) and 3rd (75th percentile) quartile, whiskers represent values between ±1.5 * interquantile range (IQR), observations outsite over ±1.5*IQR are reported as outliers (black dots); violin shapes represent the variable distribution in each sample group. Figs B-C-D: N = 9, histograms represent the average ein each group and errobars the confidence interval for 95% of observations, single observation are jittered as small dots inside each histogram. Letters indicate significance levels: a: p>0.05; b: p<0.05; c: p<0.01; d p<0.001.

(varied between 5% to over 22% of TLE) and TG (16–30% of TLE). Due to the lower relative abundance of TG, higher relative amounts of polar lipids were observed in these groups (**S3 Table**, p<0.05). Neutral lipids dominated the relative composition in ISO, OUT, CYL and NANNO, which clustered closely on the nMDS plot. In all these groups, the main lipid class was triacylglycerols (TG), which accounted for over 75% of TLE in these samples (**Fig 4B**). The predominance of TG in the previous diet groups, consequently affected the relative quantification of the polar lipid classes, resulting in significant differences when compared with T0 (p<0.001). Lastly, MONO clustered isolated from the other samples, characterised by the lowest amount of TG, two unidentified lipids $Unkn_{11.80}$ (p<0.05) and $Unkn_{12.05}$ (p<0.001) and by the elevae relative content in polar lipid (79.77±3.41%, **S3 Table**, p<0.05).

In absolute terms, TG resulted to be the main component of neutral lipids, and was the main class driving nMDS sample clustering and being affected by the diet treatments (**Fig 4A–4D**, p<0.05). The remaining neutral lipid classes were hardly affected by the diet treatments,

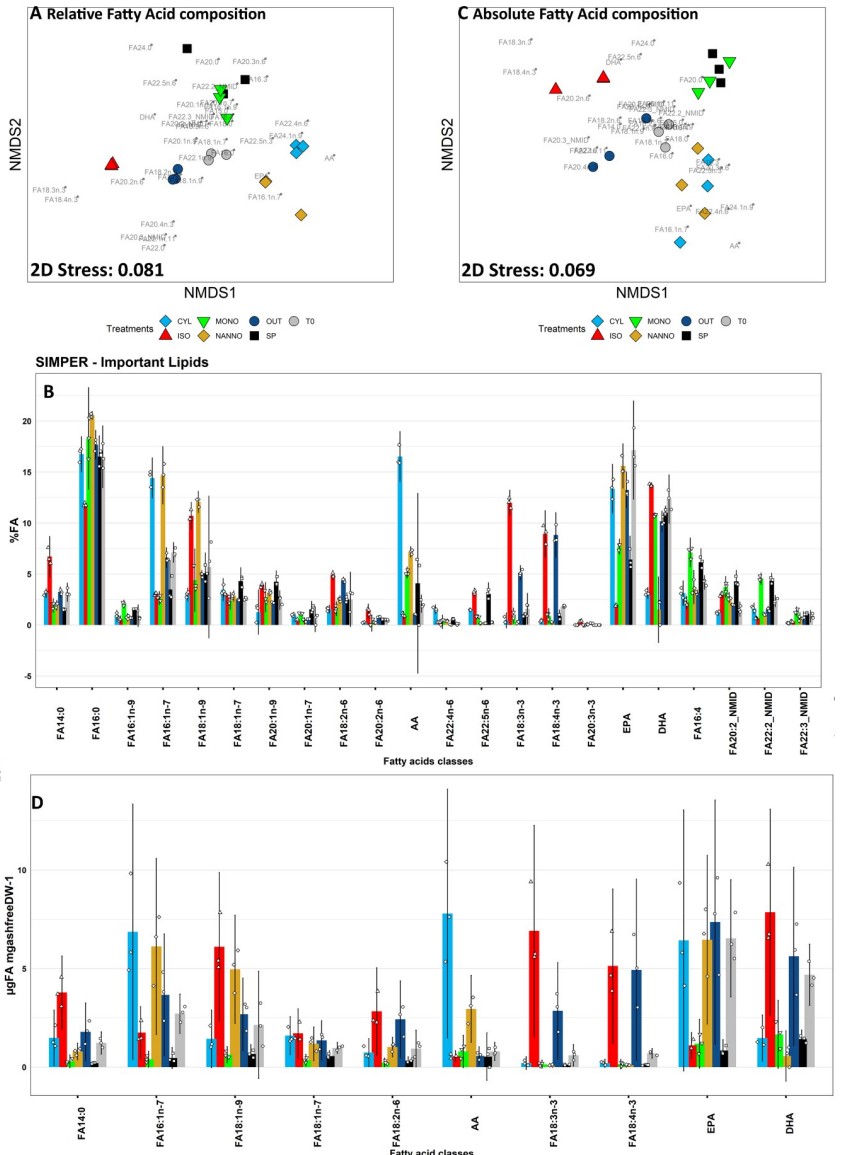

**Fig 3.** Non metric multidimensional scaling (nMDS) analisis and similarity percentages (SIMPER) analysis for relative (A-B) and absolute (C-D) fatty acid composition of spat subjected to the diet treatments. A: nMDS plot of the relative FAME composition of spat (%FA) subjected to the feeding trial. Single FA loadings are stacked on the plot and reported in grey. B: principal variables explaining for a cumulative 75% of group differences in relative spat FA composition data evidenced by SIMPER analysis. C: nMDS plot of the absolute ($\mu$gFA mg$_{ashfreeDW}^{-1}$) fatty acid composition of the spat. Single FA loadings are stacked on the plot and reported in grey. D: principal variables explaining for a cumulative 75% of group differences in absolute spat FA composition data evidenced by SIMPER analysis. T0: Spat sampled before the beginning of the trial, SP: spat fed with ShellPaste during the 4 weeks diet trial; CYL: spat fed with *C. fusiformis* 1017/2 during the 4 weeks diet trial; ISO: spat fed with *I. galbana* 927/1 during the 4 weeks diet trial; MONO: spat fed with *M. subterranean* 848/1 during the 4 weeks trial; NANNO: spat fed with *N. oceanica* 849/10 during the 4 weeks trial; OUT: spat deployed outdoor and sampled after 4 weeks. Three replicates (n = 3) for each sample group are here reported. Charts B-D: data are reported as average (histogram) ± 95% confidence interval; individual observations are jittered on the chart (smaller dots). The complete data for FA analysis of spat and statistical significance is provided in **S3 Table**.

as their absolute content remained stable between T0 and the spat diet groups. Indeed, the absolute content of polar lipids was also hardly affected by the different diets, with only LPC resulting significantly higher in MONO (p<0.001).

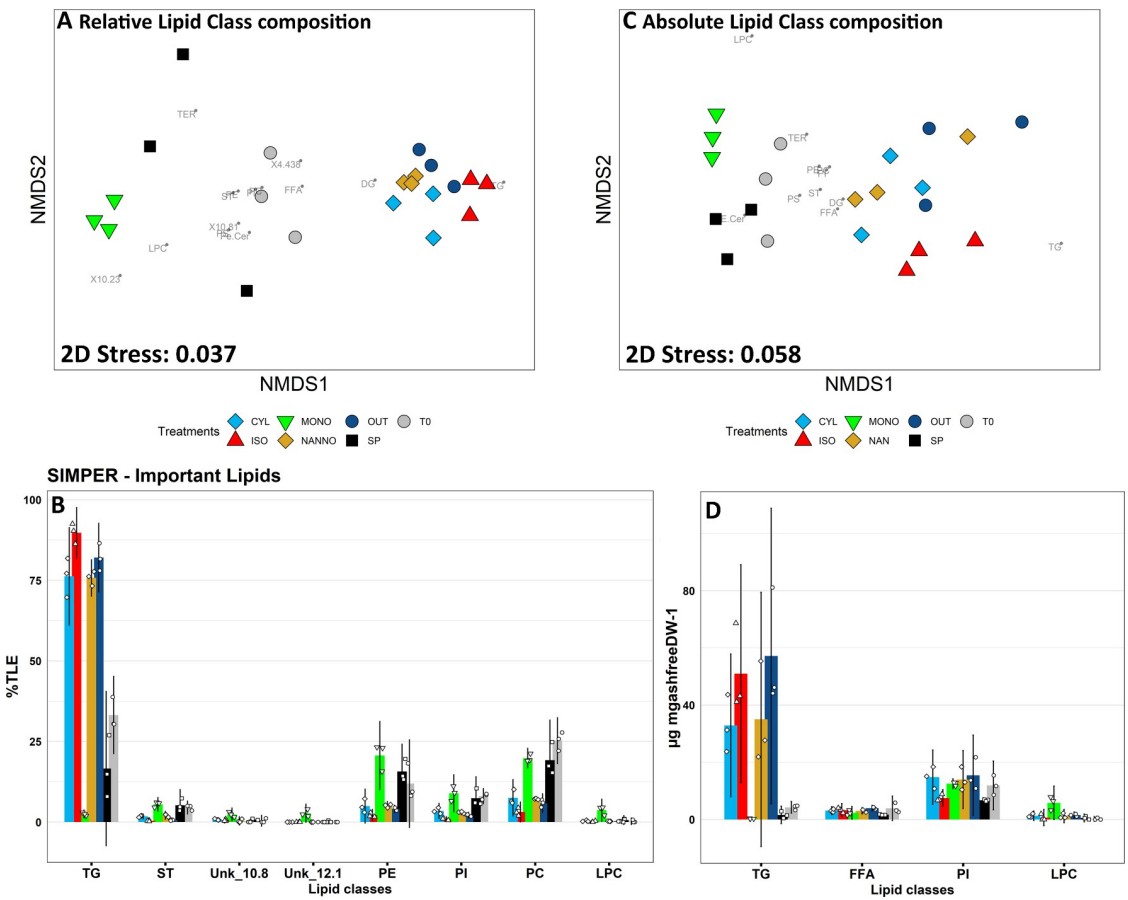

**Fig 4.** Non metric multidimensional scaling (nMDS) analisis and similarity percentages (SIMPER) analysis for relative (A-B) and absolute (C-D) lipid class composition of spat subjected to the diet treatments. A: nMDS plot for relative lipid class composition of the spat (%TLE) subjected to the feeding trial. Single lipid class loadings are stacked on the plot and reported in grey. B: principal variables explaining for a cumulative 75% of group differences in relative spat lipid class composition data evidenced by SIMPER analysis. C: nMDS plot of the absolute ($\mu$g mg$_{ashfreeDW}^{-1}$) lipid class composition of the spat. Single lipid class loadings are stacked on the plot and reported in grey. D: principal variables explaining for a cumulative 75% of group differences in absolute spat lipid class composition data evidenced by SIMPER analysis. T0: Spat sampled before the beginning of the trial; SP: spat fed with Shellfish Paste during the 4 weeks trial; CYL: spat fed with *C. fusiformis* 1017/2 during the 4 weeks diet trial; ISO: spat fed with *I. galbana* 927/1 during the 4 weeks diet trial; MONO: spat fed with *M. subterranean* 848/1 during the 4 weeks trial; NAN: spat fed with *N. oceanica* 849/10 during the 4 weeks trial; OUT: spat deployed outdoor and sampled after 4 weeks. Three replicates (n = 3) for each sample group are here reported. Charts B-D: data are reported as average (histogram) ± 95% confidence interval; individual observations are jittered on the chart (smaller dots).The complete data for lipid class analysis and statistical significance values are found in **S3 Table**.

## 3.4 Untargeted lipidomics analysis of spat

The spat lipidome exploration was completed by untargeted lipidomics. The LC-MS analysis of spat lipidome subjected to the different diets resulted in the identification of 463 features (343 of them successfully identified according to exact mass) and 620 features (267 successfully identified according to exact mass) respectively in positive (POS) and negative (NEG) ionization modes. For convenience, we will here focus on successfully identified lipid species. Plots obtained including all features are available in **S5–S7 Figs**. The LC-MS profiles and the PIT are provided in **S2 Fig** and **S1 Data**, whilst raw data are available from Mendeley data repository (DOI: 10.17632/w57zy87s68.1). POS and NEG data were separately analysed via PLS-DA, the resulting plots are shown in **Fig 5**. The different groups showed distinct lipid profiles which resulted in neat clustering of the samples. The PLS-DA score plots reported in **Fig 5A–5C** explain respectively the 77.6% and 58.5% between components 1 and 2 of PLS-DA models for

NEG and POS dataset. Cross-validation of the PLS-DA model identified in the first 4 partial least square (PLS) components the best model accuracy for POS and in 5 PLS needed for NEG (**S4B–S4D Fig**). Lipids resulting in an average VIP scores >1 are reported as average per treatment group in **Fig 5B–5D** (for the VIP scores plots for POS and NEG data please refer to **S8A and S8B Fig**). For a measure of group variability of VIP>1 lipids please refer to the histogram plots in **S8D Fig**.

In **Fig 5B** is reported the heatmap plot for the VIP>1 evidenced by PLS-DA analysis of POS spat data. Lipids observed in POS mode included mainly PC, PE and neutral lipids as cholesteryl esters (CE), Cer and TG. The analysis of main VIP of POS data resulted in the identification of seven meaningful clusters (**S8C Fig**). The main separation between sample groups in POS mode was given by the presence of TG, which were low in T0, MONO and SP (with the remaining two sample groups overlapping in POS PLS-DA plot, see **Fig 5A**). The first cluster (+C1) identified by HC, included common lipids between all the diet showing significant growth. +C1 included also low unsaturated TG, as TG(48:2)a, TG(48:1), TG(50:1) and TG (50:2) which were highly abundant in NANNO (**S8D Fig**). TG were found also in cluster 3 (+C3), which included mostly unsaturated TG (n° double bonds ≥5) principally observed in ISO and OUT, whereas TG common between ISO, OUT and CYL were found in cluster 4 (+C4). In this cluster, TG(60:13)a and TG(60:14) resulted more abundant in CYL than in OUT and ISO. Clusters 2, 5, 6 and 7 included the main PC species in each group (**S8D Fig**). PC(38:6), PC(38:7) and PC(40:7) were PC species that completed cluster 2 (+C2), which resulted abundant in ISO, T0 and OUT. Cluster 5 (+C5) included PC and PE species, characterised by 4–5 double bonds in common between CYL and NANNO. PC(36:5), which resulted the main PC in T0, NANNO and OUT (**S8D Fig**), together with PC(O-36:6/P-38:4) and PE (O-38:6/P-38:5), formed cluster 7 (+C7), composed by lipids common between T0, CYL, NANNO and OUT. The last cluster evidenced by HC analysis of POS VIP>1, resulted cluster 6 (+C6), which included lipids common between T0, MONO, SP and OUT.

The heatmap plot **Fig 5D** shows the patterns of the VIP features highlighted by PLS-DA analysis of NEG data. Polar lipids such as PE, ceramide phosphonoaminoethyl ethanolamines (CAEP), PE-Cer, PI, PS and CL were easily visualised in negative ionization mode. If compared with POS data, the patterns NEG data are less defined. HC analysis of NEG VIP>1 lipids evidenced 7 meaningful clusters (**S8C Fig**). Sample groups clustered in 3 main clusters: one including NANNO and CYL, ISO that clustered separately, and a third one including OUT, T0, SP and MONO. Lipids observed in cluster 1 (-C1) were highly abundant in CYL, and belonged to different classes, including CAEP, CL, PC, PE, PI, PS. PC(34:1), PS(O-40:3/P-40:2) and PE(O-38:3/P-38:2) formed NEG cluster 2 (-C2) and resulted highly abundant in NANNO. NEG cluster 3 (-C3) included lipids with similar intensity between CYL and NANNO. NEG clusters 4 (-C4) was formed by lipids common between T0 and OUT, including between the others PI(38:5) and PI(40:6), meanwhile, NEG cluster 5 (-C5) contained lipids common between SP, MONO, T0 and OUT. Lipids highly present in ISO formed NEG cluster 6 (-C6), which included between the others PI(40:3). Lastly, lipids principally abundant between MONO and SP, but also found in ISO, OUT and T0, constituted NEG cluster 7 (-C7).

## 3.5 Relationship between lipid composition and spat growth performances

The relationship between relevant lipids observed from FA, lipid class and lipidomics with spat GR (calculated as SI and WI) was also calculated. SI did not result in highly correlated lipids (Spearman $R^2$ <0.8) so that is not discussed here (the complete correlation tables for GR and WI are provided in **S2 Data**). On the other hand, several lipids were correlated with WI (provided in **Table 4**). The data suggest a positive correlation between accumulation of neutral

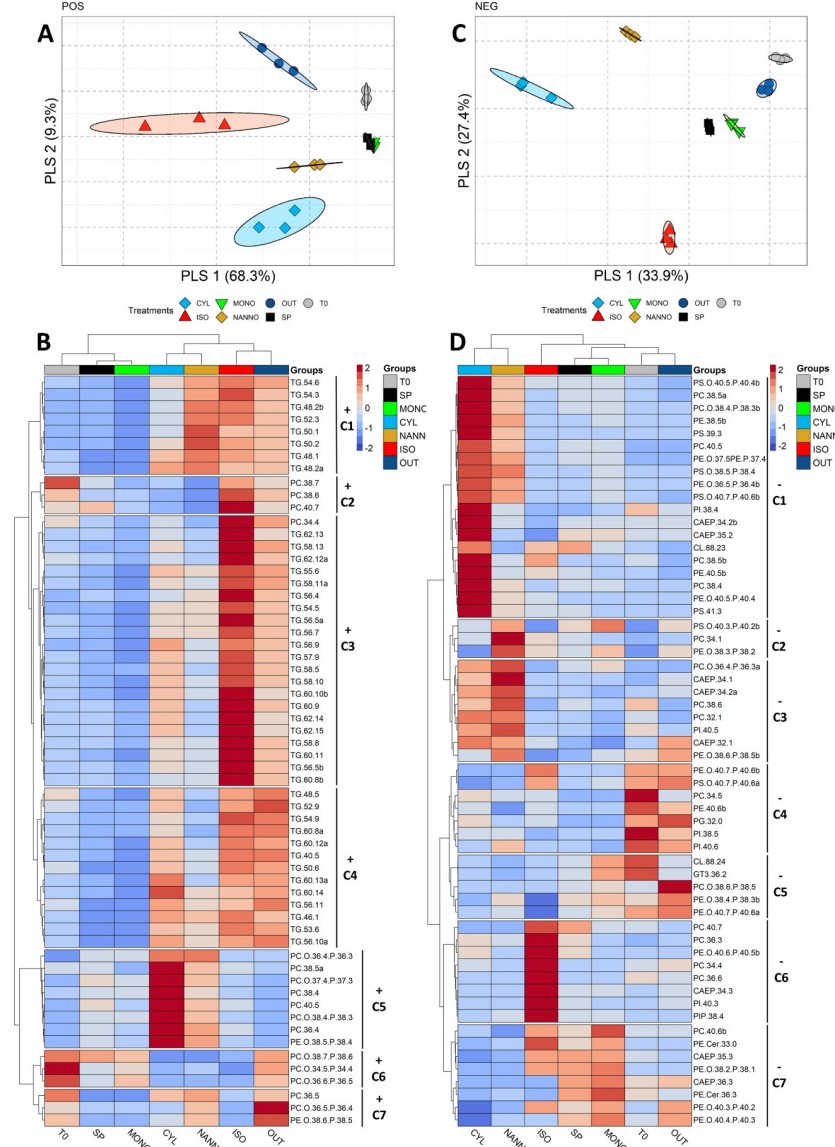

**Fig 5. Partial least squares discriminant analysis (PLS-DA) plots of untargeted lipidomics data of the spat diet groups.** A: Positive ionization mode. B: Heatmap plot reporting the group averages for Variable of important in projection (VIP) scores >1 evidenced from POS lipidomics data. C: Negative ionization mode. D: Heatmap plot reporting the group averages for Variable of important in projection (VIP) scores >1 evidenced from NEG lipidomics data. Heatmaps were plotted via 'pheatmap' package [72]. Main clusters evidenced by the elbow method (S8C Fig) are shown as breaks in the heatmap plot and are named as +/- (for POS/NEG) C and the relative cluster number (1 to 7). Euclidean distance was used as a distance measure, Ward as clustering algorithm. Lipids are reported as class, n˚ carbon and n˚ of double bonds (e.g. TG.58.10). Colour coding for lipid expression from Blue (Low) to red (High). For a representation of raw intensity for VIP selected lipids refer to S8D Fig.

lipid in spat and higher WI. TG ($R^2$ = 0.86 p<0.05) and especially for unsaturated TG species (n˚ double bonds ≥5) resulted significantly correlated with spat WI. Furthermore, also spat total content in n-3 PUFA ($R^2$ ≥ 0.88 p<0.05), total PUFA ($R^2$ ≥ 0.87 p<0.05), FA18:1n-7, FA18:2n-6 and FA20:2n-6 were significantly correlated with WI.

## 4 Discussion

For the first time to our knowledge, traditional lipid profiling techniques (as FA profiling and lipid class analysis) are here accompanied by untargeted lipidomics to approach bivalve physiology, in a comprehensive lipid analysis strategy. A large amount of information has been obtained in the past from FA analysis, however, as lipids are complex molecules with a large variety of structures and roles, many yet not fully understood in marine invertebrates [73], a comprehensive investigation is appropriate to reveal relevant information on marine invertebrate physiology [53]. The results of the three analytical approaches considered in this study offered unequivocal evidences regarding the effect of diets on mussel juveniles growth. From GR and WI analysis down to lipid molecular species, spat group could be easily classified between well performing (ISO, OUT), average performing (CYL, NANNO) and low performing (MONO, SP).

### 4.1 Growth performances: Who did grow, and who did not

Varying the microalgae composition of the diet has a profound effect on mussel spat growth performances [74]. OUT was the group characterised by the highest growth performances (SL $4.86 \pm 0.68$ mm and LW $75.08 \pm 14.2$ mg$_{LW}$spat$^{-1}$) during 4 weeks of outdoor deployment (**Fig 2**). Commonly, juvenile mussels, experience the fastest growth during summer in temperate

**Table 4. Spearman rank correlation coefficients of lipids highlighted from the dataset analysed in this study and spat live weight increase (WI).** The reported p-values are adjusted for multiple comparisons [59].

| Spearman rank correlation coefficient | | |
|---|---|---|
| **Vs Live weight increase (WI)** | | |
| **Feature ID** | **R$^2$** | **fdr adj P-value** |
| TG(40:5) | 0.94 | 2.25E$^{-05}$ |
| TG(54:9) | 0.93 | 4.60E$^{-15}$ |
| TG(50:6) | 0.93 | 4.60E$^{-15}$ |
| TG(58:8) | 0.92 | 4.60E$^{-15}$ |
| TG(56:10) | 0.92 | 4.60E$^{-15}$ |
| TG(58:11) | 0.91 | 4.60E$^{-15}$ |
| TG(52:7) | 0.91 | 4.60E$^{-15}$ |
| TG(56:5) | 0.91 | 4.60E$^{-15}$ |
| TG(52:5) | 0.91 | 4.60E$^{-15}$ |
| FA20:2n-6 | 0.90 | 4.60E$^{-15}$ |
| FA18:2n-6 | 0.89 | 4.60E$^{-15}$ |
| TG(56:7) | 0.88 | 4.60E$^{-15}$ |
| Σ n-3 | 0.88 | 4.60E$^{-15}$ |
| TG(54:5) | 0.88 | 4.60E$^{-15}$ |
| Σ PUFA | 0.87 | 4.60E$^{-15}$ |
| TG(58:10) | 0.86 | 4.60E$^{-15}$ |
| TG | 0.86 | 4.60E$^{-15}$ |
| TG(56:9) | 0.84 | 4.60E$^{-15}$ |
| TG(58:5) | 0.84 | 4.60E$^{-15}$ |
| TG(52:6) | 0.82 | 1.30E$^{-04}$ |
| TG(60:10)b | 0.81 | 2.15E$^{-04}$ |
| Σ Neutral lip. | 0.80 | 3.05E$^{-04}$ |
| FA18:1n-7 | 0.80 | 3.05E$^{-04}$ |
| TG(60:14) | 0.80 | 3.54E$^{-04}$ |

areas [22, 75]. Typical summer phytoplankton communities in the North-East Atlantic provide a variegate diet characterised by high abundance of diatoms and haptophytes, including high nutritional strains as *Chaetoceros sp.*, *Skeletonema sp.* and *Thalassiosira sp.*, which might favour mussel growth [76, 77]. The variety of microalgae available in the water column to the spat was reflected on the FA composition of OUT, characterised by large quantity of 16:1n-7 and EPA (common markers of diatoms grazing), 18:4n-3 and DHA (markers of flagellates) and 18:2n-6 which is marker of plant and microalgae detritus [78].

Considering the results obtained from SI and WI, we could classify the tested diet groups based on observed GR as: "Fast growth" for ISO and OUT, which outperformed the remaining diet treatments in term of GR; "average growth" for CYL and NANNO, which showed significant increase in SL of spat during the feeding trial; and "Low growth" SP and MONO (no growth observed). ISO, CYL and NANNO were fed to microalgae strains rich in essential PUFA, DHA for *I. galbana* and AA/EPA for *C. fusiformis* and *N. oceanica* (**Fig 1**). The observed trends agree with what was found in the past on different bivalve species, as providing either a source of DHA or EPA is known to be sufficient to meet spat nutritional requirements and obtain sustained growth [30, 32, 79]. The analysis of the FA composition of the diets suggests a preference for DHA rather than AA/EPA in mussel spat, as ISO resulted in significantly higher growth performances than the other two groups. Similar observations were made in *T. philippinarum* fed with *I. galbana*, showing faster growth compared with clams fed *T. suecica*, an EPA rich strain [80, 81]. The higher performances observed in ISO compared with CYL and NANNO diet groups could be related to the different availability of DHA between the diets. A reason for that can be found in the different role of DHA, an important component of membrane lipids [29, 82], and EPA, often catabolised and employed as an energy source [83]. DHA input could enhance membrane synthesis and growth, as bivalves have no ability to synthesize it from shorter precursors [49].

ShellPaste is a commercial alternative to live algae, which although balanced in the main essential PUFA, resulted in limited growth performances of the spat. Even lower were the growth performances in of MONO; a possible reason for this could be the dietary lack of long-chain PUFA, as also observed in similar case of bivalve juveniles subjected to diets lacking of essential PUFA [37, 79, 81, 84].

## 4.2 Fatty acid analysis: De novo synthesis of essential PUFA and accumulation of specific FA in fast growing spat

After having evaluated the effect of the diets on spat growth, which largely varied between sample groups, we concentrated the efforts in studying the diet effect on spat lipid metabolism. FA composition analysis highlighted the importance of essential PUFA supply via the diet, as mussels' spat evidenced low capabilities for *de novo* synthesis of essential n-3 PUFA; EPA and DHA have important physiological and structural roles, whilst MUFA are often catabolised or stored in reserve lipids [29, 32]. Total PUFA and total n-3 PUFA content correlated with spat WI (Spearman $R^2$>0.8 p<0.05, **Table 4**). These two parameters were depleted in slow-growing diet groups (SP, MONO), despite the similar PUFA content observed in all strains fed (**S2 Table**). This could be related with the tendency in bivalves to anabolise essential PUFA and catabolise non-essential FA as MUFAs. However, in conditions that do not ensure sufficient nutritional resources via the diet, essential PUFA are catabolised to produce energy resulting in a decreasing of n-3 PUFA and total PUFA content (as observed in MONO and SP).

DHA has a structural function in bivalves, as suggested from several authors in the past [29, 31, 32, 81–83, 85]. The results obtained in the present study show that relative DHA levels did not vary between the beginning of the trial and the diet groups ISO and OUT; whilst being

lowered in CYL, NANNO and SP (**Fig 3B**). This seems to be in accordance with what found by Caers, Coutteau [30], who observed that relative DHA content remained stable in starved or *Dunaliella tertiolecta* (species lacking essential PUFA) fed *Tapes philippinarum* spat; in the same study, DHA was instead accumulated in animals fed DHA enriched diets, whilst it decreased in spat fed EPA rich diets.

Likewise, in the present study the availability of EPA (*C. fusiformis*, *N. oceanica* and Shell-Paste) resulted in a significant reduction of DHA in the spat with sustained growth (**Fig 3B**). A reduction of DHA content was also observed in *T. philippinarum*, *Ruditapes decussatus* and *Ostrea edulis* spat subjected to EPA rich diets lacking DHA [27, 30, 42]. Da Costa [49] analysed the fatty acid assimilation in *C. gigas* larvae, observing a certain degree of elongation of EPA to 22:5n-3 in the absence of DHA. Our data suggest a different response to DHA limitation in mussel spat, as 22:5n-3 resulted low in all the sample groups (**S3 Table**). On the other hand, 22:4n-6 and 22:5n-6, observed only in traces at T0 and absent on the diets provided, were accumulated in CYL and (less) in NANNO, reaching respectively the 1.55% (0.74±0.27 µgFA $mg_{ashfreeDW}^{-1}$) and 1.49% (0.71±0.24 µgFA $mg_{ashfreeDW}^{-1}$) of TLE in CYL (**S3 Table**). Similar patterns were also observed in *O. edulis* spat subjected to an EPA and AA rich diet [42]. Elongation and *de novo* synthesis of n-3 and n-6 FA are largely dependent on the supply of shorter FA precursors, with a general rule a larger synthesis of n-3 PUFA in respect of n-6 PUFA [86]. In this case, the accumulation of AA, supplied by *C. fusiformis*, might have provided a substrate for elongation to 22:4n-6 and desaturation to 22:5n-6 rather than the elongation of other n-3 PUFA to compensate for the shortage in 22C PUFA (which was lacking from dietary inputs).

Non-methylene interrupted dienoic FA (NMID-FA) are endogenous FA characteristic of polar lipid in bivalves. Their exact role is not completely understood. They are known to be found exclusively in the polar lipids of bivalves and other invertebrates and their content, in some cases, has been found to be inversely proportional to the essential PUFA supplied by the diet [87], whilst in other cases NMID FA were considered to be only partially influenced by dietary intakes [29, 81]. Our data agree with the first hypothesis, as NMID absolute abundance increased in ISO (20:2 NMID, diet lacking of 20C PUFA, with a significant decrease of 22:2 NMID), while the relative abundance of 20:2 and 22:2 NMID FA resulted significantly higher in SP and MONO.

18C and 20C PUFA and MUFA (18:1n-9, 18:2n-6, 20:2n-6, 18:3n-3 and 18:4n-3) were higher in ISO and OUT spat groups. In the past, these FA have been observed in neutral lipid fractions of bivalve spat [29, 31, 32, 88]. The inclusion of such FA in neutral lipids could explain for the observed high correlation of 18:1n-7, 18:2n-6 and 20:2n-6 with spat WI (Spearman $R^2$ >0.8 p<0.05, **Table 4**), as neutral lipids dominated the best performing spat groups.

## 4.3 Neutral lipids, TG and energy reserves: Their content is higher in fast growing spat

The accumulation of neutral lipids (especially TG) largely discriminated groups demonstrating high growth rates from performing diet groups (see **Figs 4** and **5A and 5B**). This feature was further evidenced by the correlation observed between neutral lipids and TG content and several TG species with WI (Spearman $R^2$>0.8, p<0.05, **Table 4**); similar correlations were observed in juveniles and larvae of clams [32], mussel and scallops [89, 90]. Patterns of accumulation/depletion of neutral lipids in larvae subjected to efficient or to poor diets have been observed by previous authors on various bivalve species [29, 32, 81, 91]. Accumulation of TG in spat is connected with higher growth performances, as bivalve juveniles are known to accumulate lipid reserves during the summer to store energy for growth during the winter [25, 26, 91, 92]. Moreover, lipidomics patterns in TG species accumulation, due to the different diets,

were also evidenced (**Fig 5B**). TG species containing unsaturated FA (n° double bonds ≥5) were abundant in ISO and in OUT. These TG could be rich in 18C and 20C MUFA and PUFA, which were accumulated in ISO and OUT spat. Relatively little is understood about the FA composition of specific lipid classes in bivalve spat. Caers, Coutteau [81] observed the accumulation of 18:2n-6 and 18:1n-9 principally in neutral lipid of *C. gigas* spat, whilst Soudant, Van Ryckeghem [88] reported the accumulation of 18:2n-6, 18:4n-3 and 18:3n-3 in neutral lipids of gonads of adults scallops (*Pecten maximus*) fed with T-Iso. Relevant is the case of TG (60:13)a and TG (60:14), which were mainly observed in CYL. The elevated unsaturation and carbon content of these TG suggests the possible incorporation of long-chained PUFA, as AA which was copiously provided in *C. fusiformis* and accumulated in CYL. Long-chain PUFAs are commonly found esterified in polar lipids, however, when these are provided in excess of bivalves' nutritional requirements, these can be accumulated into neutral lipids. Indeed, when Caers, Coutteau [29] supplied an excess of DHA to *C. gigas* spat, observed an increasing percentage of this FA in the neutral lipids fraction, although on standard conditions DHA was principally found in polar lipids. Lastly, TG(48:1), TG(48:2)a, TG(50:1) and TG(50:2) were abundant in NANNO. Observing the NANNO FA profile, shorter chained MUFA were accumulated, so possibly these TG included 16:1n-7 and 18:1n-9 and SAFA as FA residuals.

## 4.4 Influence of dietary treatments on membrane and polar lipids

Lipid class composition analysis evidenced significant changes in the polar lipids $Unkn_{10.81}$, $Unkn_{12.05}$ and lysophosphocholines (LPC) between the sample groups. $Unkn_{10.81}$ and $Unkn_{12.05}$ resulted highly related to spat fed with *M. subterranean*, as these two lipids were principally observed in MONO. The polar lipids of Eustigmatales, as *Monodopsis sp.*, are known to be rich sources of glycolipids as monogalactosylglycerols (MGDG) and digalactosylglycerols (DGDG), with minor sulphoquinovosyldiacylglycerol (SQDG) [93]. The elution time-windows for these lipids, observed by previous authors applying a similar NP-HPLC separation [94, 95], could match with these unidentified lipids. Phospholipase activities in juvenile spat are lower than neutral lipases, and influenced by the diet [96]. Therefore, we could hypothesise that high content of such glycolipids, coupled with the low nutritional quality of *M. subterranean*, might have mediated the partial assimilation of them. Another polar lipid class highly present in MONO resulted LPC. LPC are products of PC metabolism, formed by the cleavage of a FA residual in position sn-1 or sn-2 by a phospholipase [97]. Increasing in LPC could be connected with the lower nutritional value of *M. subterranean* and the attempt to produce energy by polar lipid catabolism in the juvenile mussels. The catabolism of phospholipids was also observed on starved amphipods [98] and on crab larvae approaching metamorphosis [99]. Over than this, traditional lipid class composition analysis did not evidence further changes on the principal polar lipids classes (PC, PE, PI and PS), as it is indeed reported by others in the past [81].

However, working at a lipid molecular species level by untargeted lipidomics showed changes in the composition of polar lipids of the spat, offering the most complete overview of the spat lipidome. PC and PE are the main polar lipid components in bivalves [62, 64, 65, 67, 88, 100]. Bivalves have well-conserved polar lipid structures, with a SAFA (16:0 and minor 18:0) in sn-1 position and an unsaturated PUFA (EPA or DHA) on sn-2 [100]. Recently these structures have been also confirmed via LC-MS/MS on *M. edulis*, observing that the most common PC species in bivalves resulted in PC(36:5), PC(38:6) and PC(38:5) and for the plasmalogens PC(O-36:5), PC(O-38:5), PC(O-38:6), PC(P-38:5) [64]. These are mostly in agreement to what observed for T0 spat (**S8D Fig**), which was characterised by PC(36:5), PC(38:6) and their relative plasmalogens PC(O-36:5/P-36:4) and PC(O-38:6/P-38:5). PC plasmalogens were

principally observed in the OUT samples (**Fig 5B**), possibly as a result of seasonality, as an increase of plasmalogens on mussels between winter/spring and summer conditions was found by Facchini, Losito [62]. When spat were fed *I. galbana*, PC(38:6) was the most abundant PC, as this might have incorporated a DHA molecule in the sn-2 position. Lower was the content of PC(36:5), compared with T0 in this group, due to the lack of EPA obtained via the diet and observed from ISO FA profile. On the contrary, large content of PC(36:5) was observed in NANNO, as *N. oceanica* was the richest source of EPA. Whilst PC(38:4), PC(38:5) a and PC(36:4) were abundant in CYL and in a lower extent in NANNO. Both *C. fusiformis* and *N. oceanica* provided AA, which might have been incorporated in PC(36:4), whilst PC (38:4) could have included a FA22:4n-6 as PUFA and PC(38:5)a could have been constituted by FA22:5n-6 (both FA accumulated in CYL, **S3 Table**).

The NEG mode was dominated by polar lipids including PE, PI, CAEP and CL (**Fig 5C–5D**). Due to the absence of TG (as TG are not well ionised in NEG), smaller differences between the diets groups were observed. PE are the second major class of bivalves phospholipids [81, 100, 101], characterised by a small ethanolamine head, and are observed in specific domains of cell membranes as ion channels and cell-cell connections [50]. In bivalves, a large percentage of PE (around 40%) are plasmalogens [64, 65]. The analysis of NEG data suggests that the PE with the highest intensity belonged all to the plasmalogen species (**S8D Fig**). The exact role of plasmalogens in bivalves is not fully understood, some hypothesis suggests their role as membrane permeability (as commonly rich in long-chain PUFA) and adaptation to environment changes [100]. The effect of the diet treatments on PE composition was less pronounced than observed in PC, as a consequence of the large content of plasmalogens observed in this lipid class, which could not be resolved without the aid of MS/MS [102]. Nevertheless, PE abundance in the spat diet groups could be somehow connected with the different dietary inputs. Lipids with 1–2 double bonds characterised NANNO, which was the group that accumulated the largest amount of FA16:0, FA16:1n-7 and FA18:1n-9 (**S3 Table**). Whilst PE and PS with 4 and 5 double bonds were largely shared between CYL and NANNO (**Fig 5D**). PE(O-40:6/P-40:5)b was largely found in ISO and PE(38:4/P-38:3)b, PE(40:6)b, PE(O-40:7/40:6)a-b were mainly observed in T0 and OUT. SP and MONO were characterised by PE(O-38:2/P-38:1), PE(O-40:3/P-40:2) and PE(O-40:4/P-40:3).

A small number of PI were also relevant in group classification, with PI(40:5) abundant in CYL and NANNO, PI(38:4) in CYL, PI(40:3) in ISO, and PI(38:5) and PI(40:6) in T0 and OUT. PI is a substrate for phospholipase C, yielding DAG (which are directed for phospholipid synthesis [103]) and inositol-1,4,5-trisphosphate (IP$_3$), an important secondary messenger in Ca$^{2+}$ channels regulation. IP$_3$ is suspected to influence gonad maturation and fertilization in bivalves [104, 105]. The FA composition of PI can have a relevant role for physiological processes as reproduction and growth in bivalves. PI are commonly rich in AA and other long-chained PUFA, resulting in an important substrate for eicosanoids and prostaglandins production [100, 106].

Several CAEP species were also responsible for group clustering of NEG lipidomics dataset. CAEP is an important class of polar lipid in bivalves, often the third most abundant after PC and PE [107]. CAEP and the closely related PE-Cer and belong to ceramide lipids and represent for invertebrates the analogues of vertebrates' sphingomyelins. Their existence and roles were mostly unknown until the last decades when have been observed in several invertebrate species ranging from jellyfish to bivalves and insects [108, 109]. The chemical stability of CAEP, given by the sphingosine backbone and the C-P bond, making them between the most refractory components of the lipidome. The higher content of PE-Cer(33:0), PE-Cer(36:3), CAEP(35:3) and CAEP(36:3) observed in MONO and SP could be related to their resistance to

degradation and action of lipases [63]. Although several other CAEP species resulted connected with CYL, NANNO and ISO (**Fig 5D**).

CL is a further group of phospholipids, in which a third glycerol molecule is acetylated in position sn-1' and sn-2' creating a pseudo symmetrical molecule [110]. CL are predominantly located inner mitochondrial membrane where exert a relevant role in the oxidative phosphorylation process [111]. CL(88:24) and CL(88:23) were listed between the main lipids explaining for PLS-DA classification. in SP and MONO, compared with the remaining diets. Long-chain PUFA containg CL are common in bivalves, and the presence of CL(88:24) has been observed in the past in *P. maximus*, *C. gigas* and *M. edulis* [112], explaining the relatively high abundance of such lipid in T0. In bivalves, CL composition is highly conserved between species [113, 114], however, stress conditions, resulting in oxidative stress and ROS production, also influence CL composition, due to the function of this particular lipid in cellular respiration [115]. Other authors evaluated the dietary effect on mitochondrial CL, observing an increase of PUFA containing CL in mice subjected to SFA and MUFA rich diets, and on the contrary, a relative decrease of certain PUFA containing CL in presence of PUFA rich diets [116]. This trend is interestingly similar to what observed in MONO and SP, which were subjected to nutritionally poor (SP) and PUFA low (MONO) diets. Nevertheless, no final conclusion on this observation could be made, as CL(88:23) was observed in CYL and ISO (subjected respectively to EPA/AA and DHA rich diets).

## 5 Conclusions

Nursery bivalve production is the most economically demanding of bivalve hatchery practices, as the extremely delicate newly-settled juveniles suffer of high mortality when their product value is at the highest. Expanding existing knowledge of the nutritional physiology of this key life stage is essential to further develop mussel aquaculture production and for improved understanding of the ecology of this important species. Furthermore, mussel spat represent an interesting model organism, as they combine fast metabolic rates with a simple culture setup, with responses directly comparable to those of adult mussels.

The three lipid analysis techniques used concurrently herein provided valuable information on spat lipid metabolism. Classical lipid profiling techniques such as FA and lipid class analysis permitted easily to discriminate between poor-performing and better-performing spat groups. Results further supported that the supply of a source of essential PUFA (either C20 or C22) is necessary for significant growth in juvenile mussels. The application of lipidomics, over traditional lipid analysis techniques, offered the most detailed overview of the spat lipidome, allowing the study of lipids at molecular species level. Through lipidomics, the increase of TG, revealed by conventional lipid class analysis, was accompanied by profound shaping of membrane lipids composition, in relation to the dietary source of essential PUFA supply; trends that were missed by lipid class profiling. This relates to shifts in lipid molecular species abundance, rather than actual changes at lipid class level, undetected by conventional lipid class profiling.

Membrane lipids are the site of production of eicosanoids and are active in temperature acclimation mechanisms [117]. It is hypothesised that a large inbalance in this compartment could affect the acclimation ability of spat to the surrounding environment, resulting in large mortality when hatchery produced spat are deployed for rope on-growing, though further research is required to prove this hypothesis.

Overall lipidomics constituted a powerful tool to streamline and expand lipid analysis possibilities of marine organisms. The elucidation of lipids roles and metabolism in these organisms will increase our possibilities to identify their nutritional requirements and understand their responses to a changing environment.

## Supporting information

**S1 Protocol. Details of sample preparation procedures employed during spat and diet lipid extraction.**
(DOCX)

**S1 Fig. Turbidimetry data recorded before and after every feeding of the spat.**
(TIF)

**S2 Fig. Liquid chromatography mass spectrometry (LC-MS) profiles of spat total lipid extracts (TLE).** ESI POS mode profiles left traces, NEG mode profiles right traces. Data are aquired at precursor ion MS (MS') via high resolution LC-MS platform (Exactive, Thermo-Scientific). Plotted via Excalibur 4.1 (ThermoScientific).
(TIF)

**S3 Fig.** Principal Component Analysis of POS (A) and NEG (B) lipidomics spat dataset Plotted via MetaboAnalystR. CYS Cylindrotheca fusiformis fed spat, ISO Isocrysis galbana fed spat, MONO Monodopsis subterranean fed spat, NANNO Nannochloropsis oceanica fed spat, OUT Outdoor deployed spat, QC: Quality control samples, SP ShellPaste fed spat, T0: T0 samples.
(TIF)

**S4 Fig. PLS-DA analysis of spat lipidomics dataset: Model performances.** A: 2000-fold Permutation test POS data. B: 10-fold leave one out–Cross-validation analysis (LOOCV) for POS Data. $Q_2$ used as parameter of model fitting. C: 2000-fold Permutation test NEG data. B: 10 LOOCV for NEG Data. $Q_2$ used as parameter of model fitting. Plotted and calculated via MetaboAnalystR.
(TIF)

**S5 Fig.** Partial least squares discriminant analysis (PLS-DA) plots of untargeted lipidomics data acquired in POS (A) and NEG (B). Full data is used in this model including unknown features. Plotted via MetaboAnalystR.
(TIF)

**S6 Fig. Heatmap plot for the VIP>1 evidenced in PLS comp.1 by PLS-DA of spat untargeted lipidomics data in POS.** Full data is here used, including unknown features. Euclidean distance was distance measure, Ward as clustering algorithm. Lipids are rported for average in each group. Lipids are reported as class, n° carbon and n° of double bonds (e.g. TG.58.10). Colour coding for lipid expression from Blue (Low) to red (High).
(TIF)

**S7 Fig. Heatmap plot for the VIP>1 evidenced in PLS comp.1 by PLS-DA of spat untargeted lipidomics data in NEG.** Full data is here used, including unknown features. Euclidean distance was distance measure, Ward as clustering algorithm. Lipids are rported for average in each group. Lipids are reported as class, n° carbon and n° of double bonds (e.g. TG.58.10). Colour coding for lipid expression from Blue (Low) to red (High). In absence of an exact mass ID features are reported as ret. time_mass/charge (e.g. 8.86_1019.5664m/z) or ret. time_neutral mass (e.g. 11.13_1330.7461n).
(TIF)

**S8 Fig. Supplementary material employed in hierarchical clustering (HC) analysis of variable of important in projection (VIP).** A: VIP score plot resulting from PLS-DA analysis of POS dataset. B: VIP score plot resulting from PLS-DA analysis of NEG dataset. C: Evaluation

of optimal number of clusters via the "elbow method". Top POS data, Bottom: NEG data. D: Histogram plots showing the the raw intensity of VIP evidenced by PLS-DA in POS (PC POS and TG POS) and NEG data (NEG1 and NEG2). NEG1 included lipids belonging to CAEP, CL, PE-Cer and PC; NEG 2 reports lipids belonging to PE, PI and PS. VIP score plots were calculated and plotted via 'MetaboanlystR' package [1], whereas the number of meaningful HC were calculated and plotted via the R package 'factoextra' [2]. Data in D are reported as average of Normalised intensity ± SD and Plotted via Daniel's XL toolbox for Microsoft Excel.

**References S8 Fig.**

1. Xia J, Chong J. MetaboanlystR: An R package for comprehensive analysis of metabolomics data. 0.0.0.9000 ed2018.

2. Kassambara A, Mudt F. Package 'factoextra'. 1.0.5 ed2017.

(TIF)

**S1 Table. Main adducts and exact masses (for positive and negative ionization) of the lipid standard mixture used for exact mass (MS') identification of lipidomics data.**
(DOCX)

**S2 Table. Fatty acids composition of the five diets employed in this study, reported as percentage of each FAME for the total fatty acid content for each diet (% total FA) and as absolute fame content µg of FA per mg of algae dry weight (µgFA mgDW-1).** AA: arachidonic acid– 20:4n-6, EPA: eicosapentaenoic acid– 20:5n-3, DHA: docosahexaenoic acid– 22:6n-3; DMA: dimethylacetals. Data are reported as average of three replicates ± SD. Statistical differences are reported in comparison to ShellPaste. FA evidenced by SIMPER and differing significantly between diets are in **bold**. Letters correspond to statistical significance: a $p > 0.05$, b $p < 0.05$, c $p < 0.01$, d $p < 0.001$ (†).
(DOCX)

**S3 Table. Fatty acids and lipid class composition of the spat subjected to the diet treatments, reported as % of each FAME for the total fatty acid (or of the total lipid extract– TLE) and absolute content (µgFA mgashfreeDW-1 or µglipid mgashfreeDW-1 spat) for each diet group.** FAME acronyms: AA: arachidonic acid– 20:4n-6, EPA: eicosapentaenoic acid– 20:5n-3, DHA: docosahexaenoic acid– 22:6n-3; NMID: non-methylene interrupted dienoic fatty acids. Lipid class acronyms: WE: wax esters, TG: triacylglycerols, ST: free sterols, DG: diacylglycerols, FFA: free fatty acids, PE: phosphatidylethanolamines, PE-Cer: ceramide phosphoethanolamines, PI: phosphatidylinositol, PS: phosphatidylserine, PC: phosphatidylcholine, LPC: lysophosphatidylcholine. Unidentified lipid classes are reported with Unkn and their retention time span (e.g. Unkn$_{12.05}$). T0: Spat sampled before the beginning of the trial, SP: spat fed with Shellfish Paste; CYL: spat fed with C. fusiformis 1017/2; ISO: spat fed with I. galbana 927/1; MONO: spat fed with M. subterranean 848/1; NANNO: spat fed with N. oceanica 849/10; OUT: spat deployed outdoor and sampled after 4 weeks. Data are reported as the average of three replicates ± SD. Statistical significance is reported in comparison with T0. FA and lipid class evidenced by SIMPER and differing significantly between sample groups are in marked in **bold**. Letters correspond to statistical significance: a $p > 0.05$, b $p < 0.05$, c $p < 0.01$, d $p < 0.001$ (†).
(DOCX)

**S1 Data. Peak intensity tables obtained by the processing of lipidomics data via Progenesis QI software.** During data processing chromatograms were aligned to a QC samples, peak picking was done following automatic configuration of the software and inserting a threshold background noise of 1xE5 (POS) and 1xE4 (NEG). Samples are reported as normalised abundance. The reported PIT was furtherly processed via MetaboanalysR as reported in the main

text of the paper. Compounds identifiers are found on Column A. Samples are found after column G. Accepted identification codes are given according Lipid Maps and HMDB codes. Lipid ID was provided from exact MS' with Δppm <5 and relative adduct (as reported in S1 Table).

(XLSX)

**S2 Data. Spearman correlation matrix between important lipids highlighted from fatty acids (FA), lipid class and lipidomics (POS/NEG) dataset and growth rate (GR) and weight increase (WI) of spat.** Relative abundances were employed for lipidomics data, whilst absolute values were used for FA and lipid class. Correlation values were calculated via R statistical software (v 3.5).

(CSV)

# Acknowledgments

The authors declare no conflict of interest with the topic presented in this paper. We would like to thank Inverlussa Marine Services (www.inverlussa.com) for their support and spat supply and the Culture and Collection of Algae and Protozoans (CCAP) for the supply of the algae strains. The authors thanks also the Nutrition Analytical Services of the Institute of Aquaculture of the University of Stirling, Mr. James Dick for support and expertise during FAME analysis.

# Author Contributions

**Conceptualization:** Adam D. Hughes.

**Data curation:** Vincenzo Alessandro Laudicella.

**Formal analysis:** Vincenzo Alessandro Laudicella, Christine Beveridge, Nina Long, Elaine Mitchell.

**Funding acquisition:** Michele S. Stanley, Adam D. Hughes.

**Investigation:** Vincenzo Alessandro Laudicella.

**Methodology:** Mary K. Doherty, Phillip D. Whitfield.

**Project administration:** Adam D. Hughes.

**Supervision:** Stefano Carboni, Mary K. Doherty, Phillip D. Whitfield, Adam D. Hughes.

**Writing – original draft:** Vincenzo Alessandro Laudicella.

**Writing – review & editing:** Vincenzo Alessandro Laudicella, Stefano Carboni, Sofia C. Franco, Adam D. Hughes.

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
