## [Decision Letter · Decision Letter 0]

21 Oct 2019

PONE-D-19-25119

Lipidomics analysis of juveniles’ blue mussels (Mytilus edulis L. 1758), a key economic and ecological species.

PLOS ONE

Dear Mr Laudicella,

Thank you for submitting your manuscript to PLOS ONE. After careful consideration, we feel that it has merit but does not fully meet PLOS ONE’s publication criteria as it currently stands. Therefore, we invite you to submit a revised version of the manuscript that addresses the points raised during the review process.

We would appreciate receiving your revised manuscript by Dec 05 2019 11:59PM. To enhance the reproducibility of your results, we recommend that if applicable you deposit your laboratory protocols in protocols.io, where a protocol can be assigned its own identifier (DOI) such that it can be cited independently in the future. For instructions see: http://journals.plos.org/plosone/s/submission-guidelines#loc-laboratory-protocols

We look forward to receiving your revised manuscript.

Kind regards,

Juan J Loor

Academic Editor

PLOS ONE

**Journal Requirements:**

**Comments to the Author**

1. Is the manuscript technically sound, and do the data support the conclusions?

Reviewer #1: Yes

Reviewer #2: Yes

2. Has the statistical analysis been performed appropriately and rigorously? 

Reviewer #1: Yes

Reviewer #2: N/A

3. Have the authors made all data underlying the findings in their manuscript fully available?

Reviewer #1: Yes

Reviewer #2: Yes

4. Is the manuscript presented in an intelligible fashion and written in standard English?

Reviewer #1: Yes

Reviewer #2: Yes

5. Review Comments to the Author

Reviewer #1: Manuscript number: PONE-D-19-25119

Title: Lipidomics analysis of juveniles’ blue mussels (Mytilus edulis L. 1758), a key economic and ecological species

The authors represented a very good manuscript. Although the experimental results presented in this manuscript appears to be publishable, the manuscript needs to be minor revision. Therefore, it is not recommended for publication at its current state. My comments in details are as the following:

- Generally, Latin binomial must be used by authority in whole manuscript. Author name should not write italic according to Nomenclature. All author names should be added. In first stated section, species name should be written as Mytilus edulis L. and in other sections it should be written as M. edulis L. name should be written in stead of Mytilus sp.

- In abstract:

Methods should be written briefly. Also result section in the abstract should be written in an informative style. Please give real values/data, not vague subjective terms and avoid generalizations and nonessential information in the results.

Abbreviations such as GR, WI, PUFA etc. should be written clearly.

- In Material and methods section:

Abbreviations such as SAMS, FAME etc. should be written clearly.

- In result section:

Please correct “Table 3Error! Reference source not found.” in line 223 and “Error! Reference source not found.” in line 369.

- Conclusion section is very long. Authors should rewrite that part briefly.

Reviewer #2: This manuscript described some interested methods, including FA profiling, lipid class analysis and untargeted lipidomics, to evaluate of the effects of various diet on growth of mussel spat. The lipidomics analysis has potential to become a useful tool on food, aquaculture and ecological research. However, some major revision are necessary to match the publication standard of Plos One.

In “Material and Method” part, the major issues including: 1. What is the basis of the feeding trial (Why did the authors set 18 ℃ and 18:6 (L:D)for mussel feeding trial?)？ 2. Content from line 156 to 171 should be in result part. 3. The lipid extraction procedure is too sample. The authors should provided more information about it. 4. To quantify the concentrations of lipid in the algae and spat, the sample weight and water content should be measured accurately. 5. Why was 17:0 used as an internal standard? Some bacteria may have this kind FA to interfere the results. Why not use 19:0 as the standard for FA analysis. 6. It is not enough to just use known standards. It is better to identified FAMEs by GC-MS.

In the results part, all the figures are not clear enough especially Figure 5.

In disucussion part, the content in discussion 4.3 and 4.4, about TG and polar lipid is confused. I think the second paragraph in 4.3 should be in 4.4; and the second paragraph in 4.4 should be in 4.3. The authors also need to better explain the statistic results. Also, to evaluate the effect of various diet on mussel spat growth, the difference of physiological function of DHA and EPA should be considered more deeply.

The conclusion part is too long. I don’t think some content, especially the first paragraph is necessary. The results should not appear in conclusions. The authors need to give concise conclusions.

The list below are some small errors:

Line 20: “Therefore” is not necessary in abstract

Line 27: For the sentence “we applied lipidomics to bivalve nutrition”,what did the author mean

Line 34: What does GR and WI mean?

Line 116: 8 l should be 8 L, all the unit “l” in the manuscript should be “L”.

Line 117: Does L:D mean lignt : dark?

Line 119: PH should be pH

Line 119-123: What are the methods used for measuring the parameters?

Line 124: delete “.”

Lin 181: What is SL?

Line 182: What are T28 and T0

What happened in Line 222-223, 369?

Line 227: The format of the table 3 caption is different from other tables.

Line 235: I don’t think “and” should be here

Line 237: “and” should be added before “sphingosylphosphorylethanolamine”

Line 253: “Acetonitrile” should be “acetonitrile”

Line 294: What dose “1xE and 1xE” mean?

Line 323-324 ？

Line 389: “%” should be “percentage”

Line 540:The sentence “Their extract...” should be several sentences.

Line 572: Can phospholipase A cleave both sn-1 and sn-2? How about phospholipase B?

Line 577: delete one “composition”

Line 596: “%” should be “percentage”

Line 618: What is PC(38:5)a?

6. PLOS authors have the option to publish the peer review history of their article (what does this mean?). If published, this will include your full peer review and any attached files.

Reviewer #1: No

Reviewer #2: No

---

## [Author Response · Author response to Decision Letter 0]

3 Dec 2019

Dear PLOS One editorial team, 

We wish to thank you for having considered our manuscript “Lipidomics of juvenile blue mussels (Mytilus edulis L. 1758)” for publication in PLOS One. 

In this new version of the manuscript we addressed all the comments received by the reviewers. 

The abstract and conclusion sections were restructured following reviewers guidance, whereas the result section was improved as requested by Reviewer #2. The reviewer requested a clarification of all the figures in the result section, therefore figures have been replotted in larger font/contrast settings and text modified to enhance the readability and understanding of the analysis and data. 

The diet fatty acid composition is now provided in the results section. We removed the two large tables showing Fatty acid and lipid class composition of diets and spat and replaced them with more readable histogram charts including the principal variables explaining for 75% of multivariate differences between sample groups. The two large tables are still provided as supplementary material. 

In lipidomics results section, we added hierarchical clustering to connect the main lipids evidenced by PLS-DA to sample groups where these resulted more abundant. Bar-charts including raw abundance were moved as supplementary material, to provide a measure of variability of the lipidomics data. 

Other main point of reviewer #2 was the fatty acid analytical methodology, suggesting the use of different analytical platform (GC-MS instead of GC-FID) together with different internal standardization procedures. We acknowledge the relevance of the reviewer comment, however we consider GC-MS an extremely powerful method in fatty acid characterization studies, which, however does not represent the primary aim of the current manuscript. 

We employed a well-known and standardised fatty acid analysis method which included spiking FA17:0 as internal standard, and the possible pitfall of this method could be the endogenous presence of different quantities of FA17:0 in the samples, which would, in turn, represent a bias for fatty acid quantification. Correctly the reviewer pointed this out. 

To understand the endogenous presence of FA17:0 we re-analysed representative unspiked fatty acid samples via GC-FID. We observed the presence of endogenous FA17:0 in the spat tissues (1-1.5% of total FA), but not in the diet. The presence of 17:0 was also constantly 10-fold lower when compared with the corresponding samples spiked with FA17:0. Therefore, we believe that given the aim of the study, using FA17:0 could represent a legitimate approximation, as the endogenous presence of FA17:0 was constantly observed in all the spat tissues, not provided from the diet, and was not accumulated in any of the sample groups. 

The above mentioned changes are the main correction provided to the manuscript, several other smaller changes were requested by the reviewers and corrected in the new version of the manuscript, for details regarding these please consider the rebuttal letter (Responses to the reviewers) enclosed with the resubmission.

I would like to re-state the equal contribution of all the authors to the redaction of this manuscript and all of them are in agree with the content presented. We have no conflicts of interest to disclose. The raw dataset will be available through a data repository in case of manuscript acceptance.

Please address all correspondence concerning this manuscript to me at alessandro.laudicella@sams.ac.uk.

Thank you for your consideration of this manuscript. 

Sincerely,

Vincenzo Alessandro Laudicella

Response to the reviewers:

Review Comments to the Author

Reviewer #1: Manuscript number: PONE-D-19-25119

Title: Lipidomics analysis of juveniles’ blue mussels (Mytilus edulis L. 1758), a key economic and ecological species

The authors represented a very good manuscript. Although the experimental results presented in this manuscript appears to be publishable, the manuscript needs to be minor revision. Therefore, it is not recommended for publication at its current state. My comments in details are as the following:

- Generally, Latin binomial must be used by authority in whole manuscript. Author name should not write italic according to Nomenclature. All author names should be added. In first stated section, species name should be written as Mytilus edulis L. and in other sections it should be written as M. edulis L. name should be written in stead of Mytilus sp.

R.Thanks for the reviewer comment, we revised all the species names and changed accordingly. Mytilus sp is no longer reported in the manuscript. Now we reported the full scientific name at the beginning of the manuscript and then the species is reported as M. edulis. 

- In abstract:

Methods should be written briefly. Also result section in the abstract should be written in an informative style. Please give real values/data, not vague subjective terms and avoid generalizations and nonessential information in the results.

Abbreviations such as GR, WI, PUFA etc. should be written clearly.

R.Thanks for the reviewer comment. We modified the abstract paying attention to the acronyms and abbreviations. Some of the real values/data are now reported in the abstract. However, we could not report all the values for each dataset and sample group due to the limited amount of space available for the abstract section (max 300 words).

- In Material and methods section:

Abbreviations such as SAMS, FAME etc. should be written clearly.

R.Thanks for the reviewer comment. SAMS acronym is now reported as complete name. The position of the paragraph regarding diets fatty acid composition has now been moved in the result section, therefore FAME acronym is explained before being mentioned in the text.

- In result section:

Please correct “Table 3Error! Reference source not found.” in line 223 and “Error! Reference source not found.” in line 369.

R.Thanks for the reviewer for the comment. We corrected the two errors in the cross-referenced objects. 

- Conclusion section is very long. Authors should rewrite that part briefly.

R.We thank the reviewer for his comment, the conclusion section has been revised and text reduced.

Reviewer #2: This manuscript described some interested methods, including FA profiling, lipid class analysis and untargeted lipidomics, to evaluate of the effects of various diet on growth of mussel spat. The lipidomics analysis has potential to become a useful tool on food, aquaculture and ecological research. However, some major revision are necessary to match the publication standard of Plos One.

In “Material and Method” part, the major issues including: 

1. What is the basis of the feeding trial (Why did the authors set 18 ℃ and 18:6 (L:D)for mussel feeding trial?)？

R.We thank the reviewer for this comment. The two parameters were chosen based on previous experience and published litrature. 18 ⁰C was set as rearing temperature as reported in the manual for bivalve hatchery operations (Helm and Bourne, 2004). Furthermore, 18 ⁰C was the rearing temperature used also with Mytilus galloprovincialis spat (Nevejan et al., 2007) and similar temperature regimes were employed with same setup in previous trials on oysters (Carboni et al., 2016). Regarding the photoperiod, 18:6 L:D was set as normal photoperiod observed in summer in temperate areas. 

2. Content from line 156 to 171 should be in result part. 

R.We thank the reviewer for the comment and we modified the text accordingly.

3. The lipid extraction procedure is too sample. The authors should provided more information about it

R.We thank the reviewer for this comment, to extract lipids we employed the standard Folch extraction protocol, which is widely reported and known through the literature. For this reason and in an attempt to reduce the text, detailed passages of lipid extraction have been omitted. These are now included and detailed as a supplementary material (Protocol S1) in the new version of the manuscript.

4. To quantify the concentrations of lipid in the algae and spat, the sample weight and water content should be measured accurately. 

R.We thank the reviewer for this comment. No water was present in the samples as spat and algae were freeze-dried until constant weight (measured with a 0.01 mg precision scale) before lipid extraction. Please note that similar procedure have been used in the past (Albentosa et al., 1996).

5. Why was 17:0 used as an internal standard? Some bacteria may have this kind FA to interfere the results. Why not use 19:0 as the standard for FA analysis. 

R.We thank the reviewer for this comment, the 17:0 was employed as internal standard (IS) as a common procedure on fatty acid composition analysis (please see also https://www.sis.se/api/document/preview/918926/). We considered also that certain bacteria (Gram – and Mycobacteria) can synthesize 19:0 (Li et al., 2010, Perry et al., 1979), therefore, employing that as IS would still constitute an approximation. A further reason we did not employ 19:0 links with a tendency of this FA to co-elute with 18:3n-6 (Parrish, 2013). 

Nevertheless, we have undertaken additional analysis and we run spat tissues and diet FAME samples without IS to check the endogenous abundance of FA17:0. The results of the analysis confirm the absence of 17:0 inputs from the diets. On the other hand, the re-analysis of representative spat samples evidenced a relatively constant presence of endogenous 17:0 at ten-fold lower intensity (10% of the peak area of the IS peak) between all the sample groups. The text of the manuscript was modified to include this information.

We, therefore, thank again the reviewer to flag this possible flaw on the methodology employed in this study. However, it has to be considered also the scope of this study and the above-mentioned limitations in the possibility to use a different FA as IS. As the scope of the paper lies on the comparison between the effects of the different diets on the fatty acid composition of the spat, we consider the use of 17:0 as a good compromise. 

6. It is not enough to just use known standards. It is better to identified FAMEs by GC-MS.

R.We thank the reviewer for his comment, we did employ GC-FID as it result a simpler method for fatty acid quantification, as it requires the spiking of the sample with a single internal standard. Nevertheless, mix of known standards (confirmed by CG-MS) is regularly injected through the GC to ensure replicability in the retention time of all known FAs. Furthermore, unknown peaks, and lipids not reported on commercial standard mix (e.g. Dimethylacetals and NMID) were analysed also via GC-MS for identification. The text has been modified to include this information. 

In the results part, all the figures are not clear enough especially Figure 5.

R.We thank the reviewer for his comment. result section was modified to clarify the figures (which have been re-plotted with higher contrast settings). Heatmap plots have also been re-plotted increasing the readability of lipid identifiers. The two fatty acid composition tables present in the previous version of the manuscript are now reported as supplementary material and have been replaced in the text by histogram plots showing only the SIMPER evidenced lipids. Lipidomics results section has been modified and results are more thoughtfully explained. Hierarchical clustering analysis used to link different VIP features to sample groups is now commented in the result section. 

In disucussion part, the content in discussion 4.3 and 4.4, about TG and polar lipid is confused. I think the second paragraph in 4.3 should be in 4.4; and the second paragraph in 4.4 should be in 4.3. The authors also need to better explain the statistic results. Also, to evaluate the effect of various diet on mussel spat growth, the difference of physiological function of DHA and EPA should be considered more deeply.

R.We thank the reviewer for his comment. The two sections have been moved and the discussion section restructured to discuss data at a lipid class level rather than focusing on techniques employed. A section referring to the different roles between DHA and EPA on bivalves has been included in section 4.1.

The conclusion part is too long. I don’t think some content, especially the first paragraph is necessary. The results should not appear in conclusions. The authors need to give concise conclusions.

R.We thank the reviewer for his comment, the conclusion section has been revised, text reduced, and results have been removed from this section. 

the list below are some small errors:

Line 20: “Therefore” is not necessary in abstract

R.We thank the reviewer for his comment, The abstract has been modified. The word “Therefore” is no longer present. 

Line 27: For the sentence “we applied lipidomics to bivalve nutrition”,what did the author mean

R.We thank the reviewer for his comment, this sentence has been removed after restructuring of the abstract.

Line 34: What does GR and WI mean?

R.We thank the reviewer for his comment, GR abbreviation for growth rate, WI stays for weight increase. Text has been changed. 

Line 116: 8 l should be 8 L, all the unit “l” in the manuscript should be “L”.

R.We thank the reviewer for his comment, text has been changed accordingly

Line 117: Does L:D mean lignt : dark?

R.We thank the reviewer for his comment, Yes, L:D has been explained in the text. 

Line 119: PH should be pH

R.We thank the reviewer for his comment, text has been changed accordingly

Line 119-123: What are the methods used for measuring the parameters?

R.We thank the reviewer for his comment, the water parameters were measured with the instruments as reported in text.

Line 124: delete “.”

R.We thank the reviewer for his comment, text has been changed accordingly

Lin 181: What is SL?

R.We thank the reviewer for his comment, SL has been clarified in the text.

Line 182: What are T28 and T0

R.We thank the reviewer for his comment, the two terms has been clarified in the text

What happened in Line 222-223, 369?

R.We thank the reviewer for his comment. We believe that something happened during the document uploading process, as in the original word file that was uploaded on the journal website the two cross-reference objects were correctly displayed. We recreated the two captions and we hope that the problem is now resolved. 

Line 227: The format of the table 3 caption is different from other tables.

R.We thank the reviewer for spot this out, the format of table three has been changed 

Line 235: I don’t think “and” should be here

R.We thank the reviewer for spot this out, text changed accordingly

Line 237: “and” should be added before “sphingosylphosphorylethanolamine”

R.We thank the reviewer for spot this out, text changed accordingly

Line 253: “Acetonitrile” should be “acetonitrile”

R.We thank the reviewer for spot this out, text changed accordingly

Line 294: What dose “1xE and 1xE” mean?

R.We thank the reviewer for this comment, 1xE5 and 1xE4 is the intensity of the baseline noise level respectively for POS and NEG ionization mode. These two intensity levels were inserted in the software settings to reduce the occurrence of false positive compounds. The text has been modified to smooth the sentence.

Line 323-324 ？

R.We thank the reviewer for this comment, We are not sure on what the reviewer referred to. Nevertheless, as suggested above the text referring to the microalgae diet composition has been now moved to the result section, clarifying the text in these two line.

Line 389: “%” should be “percentage”

R.We thank the reviewer for spot this out, text changed accordingly

Line 540:The sentence “Their extract...” should be several sentences.

R.We thank the reviewer for this comment, but we do apologies for not being able to modify the text accordingly as it is unclear what this comment is referring to.

Line 572: Can phospholipase A cleave both sn-1 and sn-2? How about phospholipase B?

R.We thank the reviewer for this comment, we corrected this error in the text.

Line 577: delete one “composition”

R.We thank the reviewer for spot this out, text changed accordingly

Line 596: “%” should be “percentage”

R.We thank the reviewer for spot this out, text changed accordingly

Line 618: What is PC(38:5)a?

R.We thank the reviewer for this comment. In this study different isobaric species for lipids have been observed. These are lipids separated by reverse phase columns which elute between 20 and 30 seconds between each other due to differences in their acyl residuals (Please check enclosed figure for further details). However, their exact mass is identical, as their total number of carbons and double bonds does not change. To distinguish between them we employed letters after the lipid identifier (a,b,c). Explanation for the use of such nomenclature is reported in lines 247-249. 

For an image lease refer to the image attached in reviewers response file

---

## [Decision Letter · Decision Letter 1]

9 Jan 2020

PONE-D-19-25119R1

Lipidomics analysis of juveniles’ blue mussels (Mytilus edulis L. 1758), a key economic and ecological species.

PLOS ONE

Dear Mr Laudicella,

Thank you for submitting your manuscript to PLOS ONE. After careful consideration, we feel that it has merit but does not fully meet PLOS ONE’s publication criteria as it currently stands. Therefore, we invite you to submit a revised version of the manuscript that addresses the points raised during the review process.

PLEASE ADDRESS THE REMAINING MINOR ISSUES BEFORE I CAN MAKE A FINAL DECISION.

We would appreciate receiving your revised manuscript by Feb 23 2020 11:59PM. To enhance the reproducibility of your results, we recommend that if applicable you deposit your laboratory protocols in protocols.io, where a protocol can be assigned its own identifier (DOI) such that it can be cited independently in the future. For instructions see: http://journals.plos.org/plosone/s/submission-guidelines#loc-laboratory-protocols

We look forward to receiving your revised manuscript.

Kind regards,

Juan J Loor

Academic Editor

PLOS ONE

Reviewers' comments:

Reviewer's Responses to Questions

**Comments to the Author**

1. If the authors have adequately addressed your comments raised in a previous round of review and you feel that this manuscript is now acceptable for publication, you may indicate that here to bypass the “Comments to the Author” section, enter your conflict of interest statement in the “Confidential to Editor” section, and submit your "Accept" recommendation.

Reviewer #1: (No Response)

Reviewer #2: All comments have been addressed

2. Is the manuscript technically sound, and do the data support the conclusions?

Reviewer #1: Yes

Reviewer #2: Yes

3. Has the statistical analysis been performed appropriately and rigorously? 

Reviewer #1: Yes

Reviewer #2: Yes

4. Have the authors made all data underlying the findings in their manuscript fully available?

Reviewer #1: Yes

Reviewer #2: Yes

5. Is the manuscript presented in an intelligible fashion and written in standard English?

Reviewer #1: Yes

Reviewer #2: (No Response)

6. Review Comments to the Author

Reviewer #1: The author has revised the manuscript according to all recommended corrections. The manuscript is acceptable.

Reviewer #2: The authors provided well responses for my comments and suggestions. Now I think the revised manuscript is acceptable for publication. The conlusion is still too long, please compress the content again. Some errors need to corrected, for example:

Line 26-28, I think the subject was missed in the sentence.

From line 33，the are several subscript "ashfreeDW". What is it mean?

line 54: "nutrients’ bioassimilation" should be "bioassimilation of nutrients"

Line 143: "ml" should be "mL”. All "ml" should be "mL” , "ul" should be "uL" in the whole manuscript.

Line 186: what is the brand and model of the GC?

line 206: There are some errors in this line.

line 238, 310, 312: table 3 should be in one page.

7. PLOS authors have the option to publish the peer review history of their article (what does this mean?). If published, this will include your full peer review and any attached files.

Reviewer #1: No

Reviewer #2: No

---

## [Author Response · Author response to Decision Letter 1]

27 Jan 2020

Reviewer #2: The authors provided well responses for my comments and suggestions. Now I think the revised manuscript is acceptable for publication. The conlusion is still too long, please compress the content again. 

We thank the reviewer for this comment; the conclusions were compressed and text reduced as requested. 

Some errors need to corrected, for example:

Line 26-28, I think the subject was missed in the sentence.

-We thank the reviewer for this comment; the text was amended accordingly.

From line 33，the are several subscript "ashfreeDW". What is it mean?

-We thank the reviewer for this comment; ashfreeDW is the measure unit used throughout the study for absolute quantification of lipids. ashfreeDW refers to ash free dry weight (DW). The method to calculate the ash free DW is reported in section 2.2 and the acronym DW is now fully explained in the abstract. 

line 54: "nutrients’ bioassimilation" should be "bioassimilation of nutrients"

-We thank the reviewer for this comment; the text has been changed accordingly.

Line 143: "ml" should be "mL”. All "ml" should be "mL” , "ul" should be "uL" in the whole manuscript.

- We thank the reviewer for spotting this out; all unit of measure have been corrected. 

Line 186: what is the brand and model of the GC?

- We thank the reviewer for this comment; text was modified accordingly

line 206: There are some errors in this line.

- We thank the reviewer for this comment; the text has been corrected.

line 238, 310, 312: table 3 should be in one page.

- We thank the reviewer for this comment, on our word processor table 3 is contained all at the bottom of one page. The journal submission guidelines requested to locate the table directly in the main manuscript text, in the paragraph right after the first in-text citation.

---

## [Editor Report · Decision Letter 2]

29 Jan 2020

Lipidomics analysis of juveniles’ blue mussels (Mytilus edulis L. 1758), a key economic and ecological species.

PONE-D-19-25119R2

Dear Dr. Laudicella,

We are pleased to inform you that your manuscript has been judged scientifically suitable for publication and will be formally accepted for publication once it complies with all outstanding technical requirements.

With kind regards,

Juan J Loor

Academic Editor

PLOS ONE
---

## [Editor Report · Acceptance letter]

4 Feb 2020

PONE-D-19-25119R2 

Lipidomics analysis of juveniles’ blue mussels (Mytilus edulis L. 1758), a key economic and ecological species. 

Dear Dr. Laudicella:

I am pleased to inform you that your manuscript has been deemed suitable for publication in PLOS ONE. Congratulations! Your manuscript is now with our production department. 

With kind regards,

on behalf of

Dr. Juan J Loor 

Academic Editor

PLOS ONE